# Electronic metal–support interaction modulates single-atom platinum catalysis for hydrogen evolution reaction

Yi Shi [1,8 ✉], Zhi-Rui Ma[1,8], Yi-Ying Xiao[1], Yun-Chao Yin[2], Wen-Mao Huang[3], Zhi-Chao Huang[3], Yun-Zhe Zheng[4], Fang-Ya Mu[5], Rong Huang[4], Guo-Yue Shi[5], Yi-Yang Sun[6], Xing-Hua Xia [2 ✉] & Wei Chen [1,3,7 ✉]

Tuning metal–support interaction has been considered as an effective approach to modulate the electronic structure and catalytic activity of supported metal catalysts. At the atomic level, the understanding of the structure–activity relationship still remains obscure in heterogeneous catalysis, such as the conversion of water (alkaline) or hydronium ions (acid) to hydrogen (hydrogen evolution reaction, HER). Here, we reveal that the fine control over the oxidation states of single-atom Pt catalysts through electronic metal–support interaction significantly modulates the catalytic activities in either acidic or alkaline HER. Combined with detailed spectroscopic and electrochemical characterizations, the structure–activity relationship is established by correlating the acidic/alkaline HER activity with the average oxidation state of single-atom Pt and the Pt–H/Pt–OH interaction. This study sheds light on the atomic-level mechanistic understanding of acidic and alkaline HER, and further provides guidelines for the rational design of high-performance single-atom catalysts.

[1] Department of Chemistry, National University of Singapore, Singapore, Singapore. [2] State Key Laboratory of Analytical Chemistry for Life Science, School of Chemistry and Chemical Engineering, Nanjing University, Nanjing, China. [3] Department of Physics, National University of Singapore, Singapore, Singapore. [4] Key Laboratory of Polar Materials and Devices (MOE), Department of Electronics, East China Normal University, Shanghai, China. [5] School of Chemistry and Molecular Engineering, East China Normal University, Shanghai, China. [6] State Key Laboratory of High Performance Ceramics and Superfine Microstructure, Shanghai Institute of Ceramics, Chinese Academy of Sciences, Shanghai, China. [7] Joint School of National University of Singapore and Tianjin University, International Campus of Tianjin University, Binhai New City, Fuzhou, China. [8] These authors contributed equally: Yi Shi, Zhi-Rui Ma. ✉email: sychemnju@foxmail.com; xhxia@nju.edu.cn; phycw@nus.edu.sg

Hydrogen has emerged as a green and sustainable fuel to meet the demand for future global energy[1-3]. Nowadays the majority of hydrogen is still produced from steam-reformed methane, which is derived from limited fossil resources and greatly increases $CO_2$ emission. Electrocatalytic hydrogen evolution reaction (HER) enabled by renewable electricity holds great promise as a safe, scalable, low-cost, and environmental-friendly pathway for hydrogen production[4-6]. To date, noble metals (e.g., Pt, Pd, and Rh) are regarded as the most efficient materials to catalyze the conversion of $H_3O^+$ (acid) and $H_2O$ (alkaline) to $H_2$[7]. In order to maximize the utilization efficiency of noble metals, the rational design and controllable synthesis of catalysts based on the deep understanding of reaction mechanism and structure–activity relationship is crucial for cost-efficient HER catalytic process[8,9]. An effective approach for mechanistic study of the structure–activity relationship is to modulate the electronic structure of catalysts and unravel the factors that govern their catalytic activities[8].

Several strategies—multimetallic construction that integrates metal components with distinctive electronic properties, surface engineering of metal by organic modifiers, and metal–support interaction modulation—have been developed to tune the electronic structure of metal catalysts[8,10-12]. In industrial heterogeneous catalysis, metal nanoparticles are immobilized on a support and the electronic structure of the active sites on metal nanocatalysts can be effectively regulated through the strong metal–support interactions, which is rationalized as the electronic metal–support interaction (EMSI) proposed by Rodriguez and colleagues[13,14]. EMSI is associated with the orbital rehybridization and charge transfer across the metal–support interface, leading to the formation of new chemical bonds and the realignment of molecular energy levels[15-17]. The electron transfer modulates the $d$-band structure of metal nanocatalysts, strengthens the adsorption of reaction intermediates, and hence lowers the energy barrier and facilitates the rate-limiting step. However, the rearrangement of electrons with considerable EMSI effect is only confined to a couple of atomic layers at the metal–support interface[11]. For the conventional supported metal nanocatalysts consisting of few atomic layers near the interface and non-uniform atomic coordination sites (e.g., vacancies, step edges, kinks and corner sites)[18], it is difficult to correlate the overall catalytic activity with the EMSI effect on the electronic structure of active sites.

Single-atom metal catalyst (SAMC) minimizes the structure of metal components with the well-defined active sites, which are located at the metal–support interface and exposed to reactive species[19,20]. The homogeneous atomic coordination environment makes SAMC an ideal and simplified model system for the mechanistic investigation of catalytic reactions. Strong EMSI not only stabilizes the single-atom metals owing to the formation of thermodynamically favorable metal–support bonds, but also leads to the charge redistribution induced via electron transfer[21-24]. The net electron transfer from the single-atom metals to the electronegative atoms (e.g., C, N, O, S) on support positively charges the metal atoms with high oxidation state, and thus modulates the $d$ state of single-atom metals[21-27]. The $d$-orbital electrons of transition metal atoms participate directly in the catalytic redox reactions, and thus have close relationship with the adsorption strength of the catalytic reaction intermediates[20,28-31]. Although SAMC has been widely used for catalyzing HER, comprehensive atomic-level insights into the structure–activity relationship of SAMC for a wide-pH-range HER are rarely reported.

Transition metal dichalcogenides (TMDs) have been widely used as the supports for immobilizing SAMC in heterogeneous catalysis[32-37]. Compared with SAMC supported on carbon-based materials[38-40], the electronic structure of single-atom metals supported on TMDs is usually adjusted by both the anchoring atom and the neighboring transition metal atoms with relatively high atomic number, which affords a more flexible and complex coordination environment to regulate the catalytic activity[24,41]. Owing to the various well-defined band structures of TMDs (e.g., $MoS_2$, $WS_2$, $MoSe_2$, and $WSe_2$, Fig. 1a)[42], the core anchoring chalcogen (S, Se) and the neighboring transition metal (Mo, W) can synergistically regulate the electronic structure of SAMC through EMSI. The tuneable $d$-orbital state of single-atom Pt changes the adsorption energy of reactants on metal atoms and thus influences the catalytic activity of HER (Fig. 1a).

Herein, we used the previously reported site-specific electro-deposition technique[43] to construct four kinds of single-atom Pt catalysts on different two-dimensional TMDs supports ($MoS_2$, $WS_2$, $MoSe_2$, and $WSe_2$) as a model system. Detailed spectroscopic and electrochemical characterizations show that the fine tailoring of the oxidation state of single-atom Pt through EMSI activates the alkaline and acidic HER (mass activity) up to 73-fold (Pt-SAs/$MoSe_2$) and 43-fold (Pt-SAs/$WS_2$) higher than that of the commercial Pt/C, respectively, revealing the universality of the single-atom Pt system for the wide-pH-range HER investigations. With the decrease in oxidation state of single-atom Pt, the hydrogen binding energy decreases, and consequently the acidic HER activity increases to a record level. In the alkaline HER, the single-atom Pt catalyst with optimal oxidation state (ca. +2)—showing neither too strong catalyst–H interaction for hydrogen release, nor too weak catalyst–OH interaction for water dissociation—exhibits exceptional catalytic activity.

## Results

**Synthesis and characterization of single-atom Pt catalysts**. For the self-terminating growth of single-atom Pt on various TMDs supports (Fig. 1b), Cu atoms were first underpotentially deposited on the chemically exfoliated TMDs (ce-TMDs, Supplementary Figs. 1–3). The Cu atoms were then galvanically exchanged by Pt (II) (Supplementary Figs. 4 and 5), forming TMDs-supported single-atom Pt samples (denoted as Pt-SAs/TMDs: Pt-SAs/$MoS_2$, Pt-SAs/$WS_2$, Pt-SAs/$MoSe_2$, and Pt-SAs/$WSe_2$). The surface-limited underpotential deposition (UPD) technique enables the formation of energetically favorable metal–support bonds, and automatically terminates the sequential formation of metallic bonding, confirming the growth of single-atom Pt[43]. The Pt loadings on ce-$MoS_2$, ce-$WS_2$, ce-$MoSe_2$, and ce-$WSe_2$ were 5.1, 4.1, 4.7, and 4.9 wt%, respectively, as revealed by inductively coupled plasma optical emission spectrometry. No Pt-containing clusters/nanoparticles or crystalline Pt phases were observed on the ce-TMDs nanosheets (Supplementary Figs. 6, 7). Owing to the discrete distribution of Pt atoms, electrochemical cyclic voltammograms of all the Pt-SAs/TMDs samples did not show characteristic peaks of Pt in the regions of Pt-H adsorption/desorption (Supplementary Fig. 8). The similar Raman spectra of Pt-SAs/TMDs samples with those of the pure ce-TMDs imply that the metallic 1T phase of TMDs well retained after single-atom Pt decoration (Supplementary Figs. 3, 9). The aberration-corrected high-angle annular dark-field-scanning TEM (HAADF-STEM) images confirmed the atomically dispersed Pt atoms (bright spots) on the ce-TMDs nanosheets (Fig. 1c–f). The STEM-coupled energy-dispersive spectroscopy element mapping showed the homogeneous dispersion of atomic Pt over the whole samples (Fig. 1c–f).

**Electronic and coordination structure**. We then investigated the chemical configuration and local coordination of Pt-SAs/TMDs through the combination of X-ray photoelectron spectroscopy

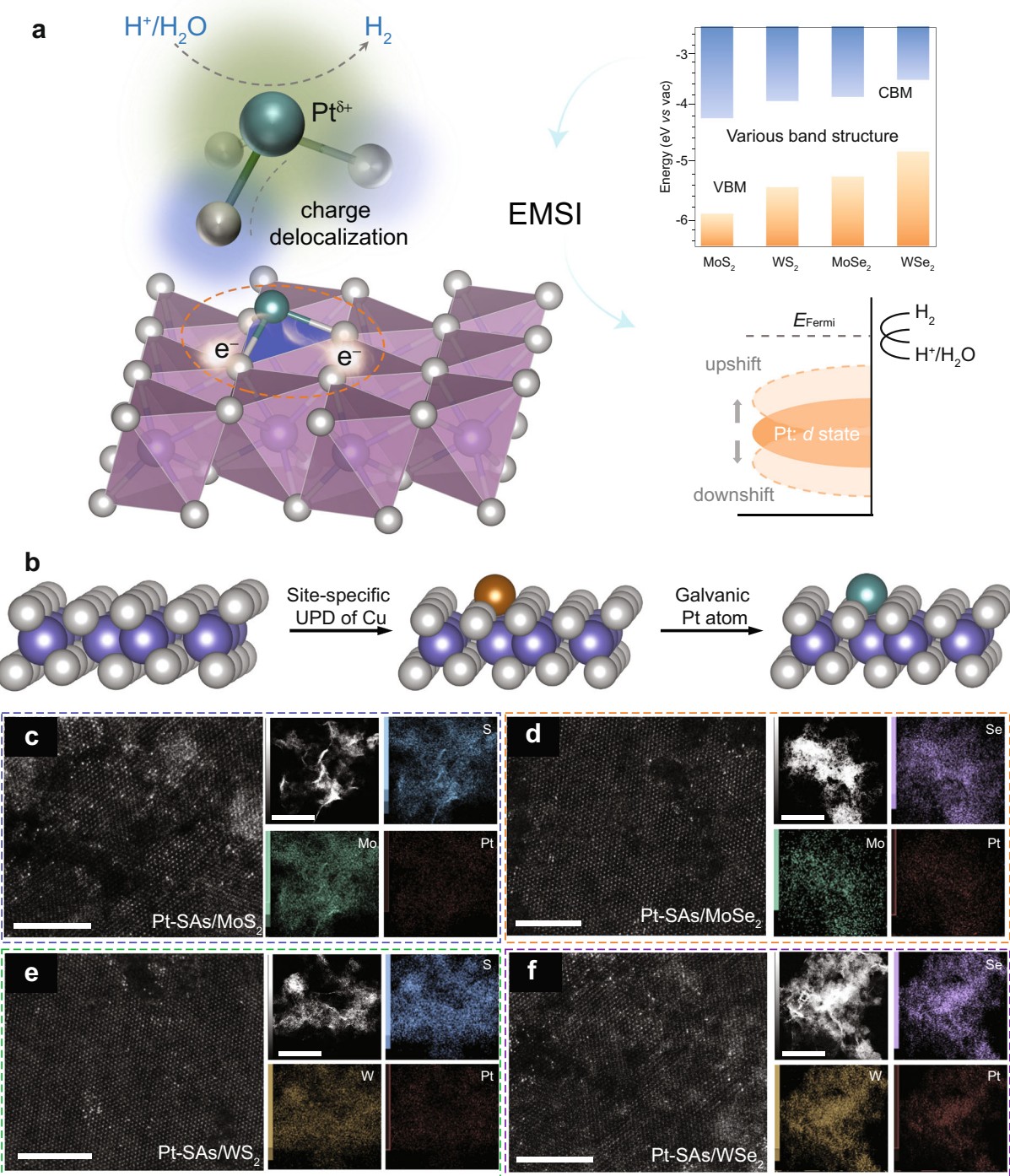

**Fig. 1 Rational design and construction of single-atom Pt catalysts. a** Electronic metal–support interactions (EMSI) modulation of single-atom Pt for catalyzing HER. Left: schematic structure of single-atom Pt on TMDs material. The gray, purple, and green spheres represent the chalcogen (sulfur/selenium), transition metal (molybdenum/tungsten), and platinum, respectively. The electronic structure of single-atom Pt was modulated by two-dimensional TMDs through charge delocalization, enabling the single-atom Pt to take slightly positive charge ($Pt^{\delta+}$). The structural unit of single-atom Pt was circled by the orange dashed line and further enlarged above. Top right: schematic diagram of the band edges of TMDs. The conduction band minimum (CBM)/valence-band maximum (VBM) band edges of TMDs (theoretical values) refer to ref. [42]. The schematic band structure—showing the electron affinity and ionization potential of various TMDs—provides a guideline for rationalizing the EMSI modulation of single-atom Pt. Bottom right: schematic illustrating that the *d*-state shift of single-atom Pt induced by EMSI regulates the catalytic performance of HER. **b** Fabrication of TMDs-supported single-atom Pt. The gray, purple, brown, and green spheres represent chalcogen (sulfur/selenium), transition metal (molybdenum/tungsten), copper, and platinum, respectively. After site-specific electrodeposition of Cu adatoms on the support, galvanic replacement of Cu adatoms with $PtCl_4^{2-}$ is carried out to produce TMDs-supported single-atom Pt. Atomic-resolution HAADF-STEM images for (**c**) Pt-SAs/MoS₂, (**d**) Pt-SAs/MoSe₂, (**e**) Pt-SAs/WS₂, (**f**) Pt-SAs/WSe₂ (scale bars: 5 nm) and the corresponding elemental mappings (right side, scale bars: 100 nm).

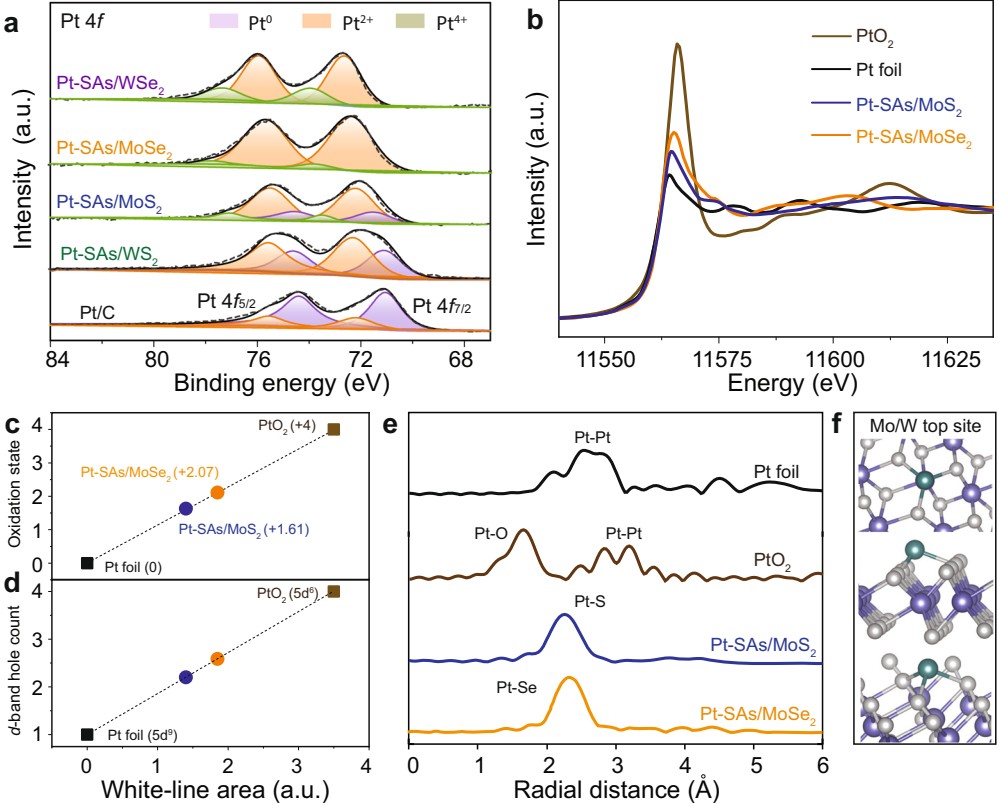

**Fig. 2 Structural characterizations of the catalysts. a** Pt 4f XPS spectra of the Pt-SAs/TMDs samples and commercial Pt/C. **b** Normalized XANES spectra at the Pt $L_3$-edge of Pt foil, PtO$_2$, Pt-SAs/MoS$_2$, and Pt-SAs/MoSe$_2$. The fitted average oxidation states (**c**) and d-band hole counts (**d**) of Pt from XANES spectra. **e** First-shell fitting of EXAFS spectra of Pt foil, PtO$_2$, Pt-SAs/MoS$_2$, and Pt-SAs/MoSe$_2$. **f** Top, side, and perspective views (up to bottom, respectively) of the geometric configurations of Pt-SAs/TMDs with the Pt atom on the Mo/W top site. The color code is the same as in Fig. 1a.

(XPS) and X-ray absorption spectroscopy (XAS). As shown by XPS in Fig. 2a, compared to that of commercial Pt/C located dominantly at 71.2 eV, the binding energies of Pt $4f_{7/2}$ of Pt-SAs/TMDs increased with obvious peak shape changes, located at approximately 71.8–72.8 eV. In order to better support the average oxidation states of Pt, the Pt$^{4+}$ and Pt$^{2+}$ references were also shown in Supplementary Fig. 10. These results demonstrated that the Pt species in Pt-SAs/TMDs were partially oxidized with 0−+4 valence state (Pt$^{\delta+}$), owing to the mutual electronic interactions between Pt atoms and ce-TMDs supports. Quantitative peak deconvolution and integration of XPS analysis showed that the average oxidation states of Pt in Pt-SAs/WS$_2$, Pt-SAs/MoS$_2$, Pt-SAs/MoSe$_2$, and Pt-SAs/WSe$_2$ were 1.24, 1.71, 2.11, and 2.61, respectively[44] (Fig. 2a, and Supplementary Table 1). The binding energies of Mo 3d/W 4f and S 2p/Se 3d of the supports decreased slightly (Supplementary Fig. 11) while the binding energy of Pt 4f increased, indicating that the electrons were transferred from Pt to ce-TMDs supports. In X-ray absorption near-edge spectroscopy (XANES), the Pt $L_3$-edge analysis (Fig. 2b) showed that the white-line intensity for Pt-SAs/MoS$_2$ and Pt-SAs/MoSe$_2$ were higher than that for Pt foil, indicating more unoccupied 5d orbitals for single-atom Pt. The oxidation states of single-atom Pt in Pt-SAs/MoS$_2$ and Pt-SAs/MoSe$_2$ samples were 1.61 and 2.07 (Fig. 2c and Supplementary Fig. 12), respectively, in line with the XPS analysis. According to Pt$^0$ foil (5d$^9$6s$^1$) and Pt$^{IV}$O$_2$ (5d$^6$6s$^0$) standards, the number of d-band hole for Pt-SAs/MoSe$_2$ was estimated to be 2.590, higher than that for Pt-SAs/MoS$_2$ (2.205), indicative of the increased d vacancy[29] (Fig. 2d). Because of the interference of W k-edge signal, the Pt L$_3$ edges of Pt-SAs/WS$_2$ and Pt-SAs/WSe$_2$ could not be well recorded[45].

To further demonstrate the homogeneous coordination environment of the model system, we further evaluated the local atomic structure of single-atom Pt by extended X-ray absorption fine structure (EXAFS), density functional theory (DFT)-optimized structural models, as well as magnified HAADF-STEM images. The EXAFS spectra of Pt-SAs/MoS$_2$ and Pt-SAs/MoSe$_2$ showed a similar peak at approximately 2.3 Å, ascribed to the Pt–S and Pt–Se bonds with coordination numbers of 3.2 and 3.5, respectively (Fig. 2e, Supplementary Figs. 13, 14, and fitting parameters shown in Supplementary Table 2)[23,43]. No appreciable Pt–Pt bond (2.8 Å) was detected in Pt-SAs/MoS$_2$ and Pt-SAs/MoSe$_2$ (Fig. 2e), further verifying the dominant existence of single-atom Pt. No observation of Pt–Mo coordination ruled out the covalent attachment of Pt atom on Mo edge site or S vacancy. Combined with the coordination numbers, the magnified HAADF-STEM images implied that Pt atoms, coordinating with three nearest neighboring S or Se, were straddled atop Mo (Supplementary Fig. 15). In line with these results, the theoretical Pt–S (2.22, 2.22, and 2.47 Å) and Pt–Se distance (2.28, 2.28, and 2.56 Å) obtained from the optimized Mo atop model (Supplementary Fig. 16) agreed well with the EXAFS analysis (average bond length 2.27 and 2.35 Å for Pt–S and Pt–Se, respectively), which further confirmed Pt attachment on the Mo top site (Fig. 2f). The high-energy barrier for Pt diffusion between the nearest adsorption sites demonstrated the structural stability of the single-atom catalysts (Supplementary Fig. 16).

**Electrochemical HER study.** The structural characterizations have shown that the electronic structure of single-atom Pt could be fine-tuned by different TMDs supports through EMSI. We then attempted to demonstrate the effect of EMSI modulation on the

catalytic performance of single-atom Pt. Considering the different HER reactants in alkaline ($H_2O$) and acidic ($H_3O^+$) media (detailed mechanisms shown in Supplementary Fig. 17)[7], we investigated the electrocatalytic HER activity of Pt-SAs/TMDs using a typical three-electrode configuration in wide-pH-range electrolytes, including 1.0 M KOH and 0.5 M $H_2SO_4$ solutions (Supplementary Fig. 18).

In the alkaline condition, all the Pt-SAs/TMDs samples showed much superior electrocatalytic HER activity with negligible overpotential, compared to the pristine ce-TMDs supports (Supplementary Fig. 19). It should be noted that these supported single-atom Pt samples exhibited various alkaline HER activities, with the order of Pt-SAs/$MoSe_2$ > Pt-SAs/$MoS_2$ > Pt-SAs/$WS_2$ > Pt-SAs/$WSe_2$ (Fig. 3a). Specifically, Pt-SAs/$MoSe_2$ displayed the overpotential value of approximately 29 mV at a current density of 10 mA $cm^{-2}$ (left axis of Fig. 3b) and exceptional mass activity of 34.4 A $mg^{-1}$ under an overpotential of 100 mV (right axis of Fig. 3b, details of calculation shown in Supplementary Note 1), which was the best among all four samples. The electrocatalytic HER activity of Pt-SAs/$MoSe_2$ was remarkably 73.4-fold higher than that of commercial Pt/C, and exceeds most of the state-of-the-art SAMCs or Pt-based electrocatalysts (Fig. 3b, Supplementary Fig. 20, and references shown in Supplementary Tables 3, 4). Gas chromatography analysis was applied to verify the catalytic production of $H_2$ (Supplementary Fig. 21).

To investigate the mechanistic insights into the alkaline HER activity of Pt-SAs/TMDs samples, we evaluated the catalysis kinetics from Tafel plots (Fig. 3c). Compared to the other Pt-SAs/TMDs samples (50, 59, and 65 mV $dec^{-1}$), Pt-SAs/$MoSe_2$ exhibited the smallest Tafel slope of 41 mV $dec^{-1}$, implying the fastest HER kinetics. Tafel slopes were much lower than 120 mV $dec^{-1}$ in all Pt-SAs/TMDs samples, suggesting that the prior sluggish Volmer reaction was greatly accelerated. The turnover frequency (TOF) value of the Pt sites on Pt-SAs/$MoSe_2$ was 6.21 $s^{-1}$ (at −50 mV; details of the calculation shown in Supplementary Note 2), which strikingly surpasses the other three Pt-SAs/TMDs samples (Fig. 3d) and previously reported single-atom electrocatalysts (references shown in Supplementary Table 3). The negligible degradation of Pt-SAs/TMDs after 1000 cycles stability tests demonstrated the high stability of the supported single-atom Pt (Supplementary Fig. 22), which is an essential prerequisite for mechanism investigation. The HAADF-STEM and XPS characterizations of Pt-SAs/TMDs after HER measurements suggested the unchanged morphology and valence state of single-atom Pt during HER, further confirming the stability of the Pt-SAs/TMDs catalysts (Supplementary Figs. 23, 24, and Supplementary Table 5).

Similarly, we also showed the various acidic HER activities on different supported single-atom Pt samples, which followed the order distinctive from that in alkaline HER: Pt-SAs/$WS_2$ > Pt-SAs/$MoS_2$ > Pt-SAs/$MoSe_2$ > Pt-SAs/$WSe_2$ (Figs. 3e and 3f), consistent with the previously reported results[43]. For instance, compared to the other three samples, Pt-SAs/$WS_2$ displayed much lower overpotential value (32 mV; left axis of Fig. 3f) at a current density of 10 mA $cm^{-2}$ and exceptional mass activity of 130.2 A $mg^{-1}$ under an overpotential of 100 mV (43.1-fold higher than commercial Pt/C; right axis of Fig. 3f, details of the calculation shown in Supplementary Note 1). The Tafel slope and TOF values of Pt-SAs/$WS_2$ were 28 mV $dec^{-1}$ and 273 $s^{-1}$ (at −200 mV; details of the calculation shown in Supplementary Note 2), respectively, much better than the other three Pt-SAs/TMDs samples and most of the state-of-the-art SAMCs or Pt-based electrocatalysts (Figs. 3g, 3h, Supplementary Fig. 25, and references shown in Supplementary Tables 6, 7).

We further verified the active sites for HER and the negligible contribution of TMDs support to HER. The thiocyanate ions ($SCN^-$) poison experiment of the Pt-SAs/TMDs samples was conducted to efficiently block the Pt sites for acidic HER[34,43].

Upon the addition of $SCN^-$, the HER current of all the Pt-SAs/TMDs samples decreased dramatically approaching near zero, confirming that HER activity dominantly derives from the single-atom Pt and the catalytic performance enhancement is mainly attributed to the EMSI modulation of Pt (Supplementary Fig. 26). This phenomenon is distinct from the Pt-doping case, where Pt atoms are incorporated into the TMD lattice and chalcogen atoms are reported as the active sites of HER[33] (Supplementary Note 3, and Supplementary Table 8). Additionally, the Tafel behavior of Pt-SAs/TMDs in acidic HER (~30 mV $dec^{-1}$) resembles that of the commercial Pt, indicating that the catalytic reaction on single-atom Pt contributed mostly to the HER. In contrast, it has been reported that Pt-doped $MoS_2$ showed a Tafel slope of 96 mV $dec^{-1}$, close to that of pure $MoS_2$ (~100 mV $dec^{-1}$)[33].

**Structure and activity relationship**. To evaluate the EMSI effect on the hydrogen adsorption ability of different single-atom Pt catalysts, the highly surface-sensitive ultraviolet photoelectron spectroscopy (UPS) was conducted to probe the occupied electronic states of single Pt atoms on different supports (Fig. 4a). For a free-state single-atom metal, no metallic bonding is formed and thus only the *d* level exists. The *p–d* orbital hybridization between single Pt atom and coordinating atom (e.g., S and Se) on the support broadens the *d* level of single-atom Pt, leading to the formation of a narrower *d* band around 0–4 eV compared to that of bulk Pt (Fig. 4a and Supplementary Fig. 27)[46–49]. The positions of *d*-band center for Pt-SAs/$WS_2$, Pt-SAs/$MoS_2$, Pt-SAs/$MoSe_2$, and Pt-SAs/$WSe_2$ were −2.58 eV, −2.39 eV, −2.24 eV, and −2.06 eV, respectively (Fig. 4a), higher than that of Pt nanocrystal (−2.94 eV)[48]. Due to the EMSI modulation, various density of states patterns were obtained from the DFT calculation for the Pt-*d* orbitals of the four Pt-SAs/TMDs (Supplementary Fig. 28). The *d*-band model developed by Norskov and colleagues has been widely used in relating adsorption properties of rate-limiting intermediates to the electronic structure of catalysts[28,50]. According to the *d*-band theory, when the Pt *d* band experiences an upward shift, more antibonding states of hydrogen are pulled above the Fermi level, and hence strengthens the affinity of Pt towards hydrogen (Fig. 4b)[28,50]. On the basis of the experimental UPS valence-band spectra (VBS), the H adsorption ability of these single-atom Pt samples can be proposed following the order: Pt-SAs/$WS_2$ < Pt-SAs/$MoS_2$ < Pt-SAs/$MoSe_2$ < Pt-SAs/$WSe_2$. This trend predicted by the *d*-band theory also agreed well with the Gibbs free energy of atomic hydrogen adsorption ($\Delta G_{H^*}$, Supplementary Fig. 29) obtained from the DFT simulation.

The correlation of electronic structure, hydrogen binding energy (HBE) and acidic HER activity is shown (Fig. 4c) with the average oxidation state of single-atom Pt (determined by XPS and XANES) as x-axis while hydrogen adsorption ability (represented by the *d*-band center from VBS) as left y-axis and the overpotential required to achieve a current density of 10 mA $cm^{-2}$ as the right y-axis. With the increase of average oxidation state of single-atom Pt, the hydrogen adsorption ability also increased in a nearly linear relation (Fig. 4c), implying the EMSI modulation on the adsorption energy of H* intermediates during HER. We showed that the HER activity of the single-atom Pt catalyst decreased monotonically with the increase of average oxidation state and H adsorption ability (Fig. 4c), thus providing solid evidence supporting that HBE is the dominant descriptor in the acidic HER of single-atom Pt. The near ambient-pressure X-ray photoelectron spectroscopy (NAP-XPS) further demonstrated that stronger hydrogen adsorption on single-atom Pt with higher valence state (Pt-SAs/$WSe_2$) leaded to active site poisoning and slow hydrogen desorption, greatly limiting the overall HER activity (Supplementary Figs. 30, 31).

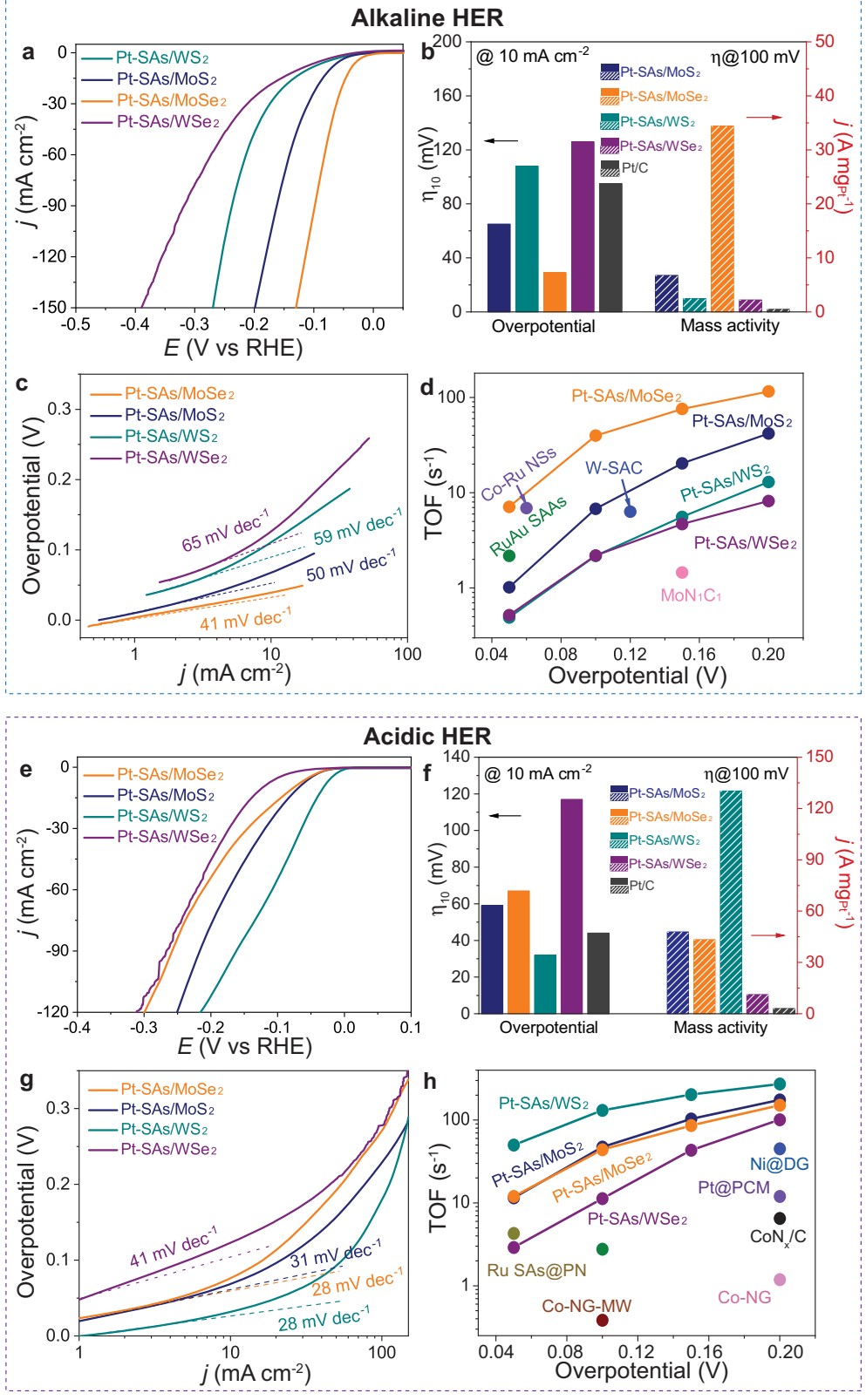

**Fig. 3 Electrochemical analysis in 1.0 M KOH and 0.5 M H₂SO₄. a**, **e** HER polarization curves of various Pt-SAs/TMDs samples. **b**, **f** HER comparison of overpotentials required to achieve a current density of 10 mA cm⁻² (black arrow, left axis) and mass activities (normalized by the Pt loading, red arrow, right axis) at −100 mV versus RHE for various Pt-SAs/TMDs samples. **c**, **g** Tafel plots derived from the early stage of the corresponding HER polarization curves. **d**, **h** Turnover frequency (TOF) curves of various Pt-SAs/TMDs samples and a comparison with the previously reported state-of-the-art values for single-atom HER catalysts.

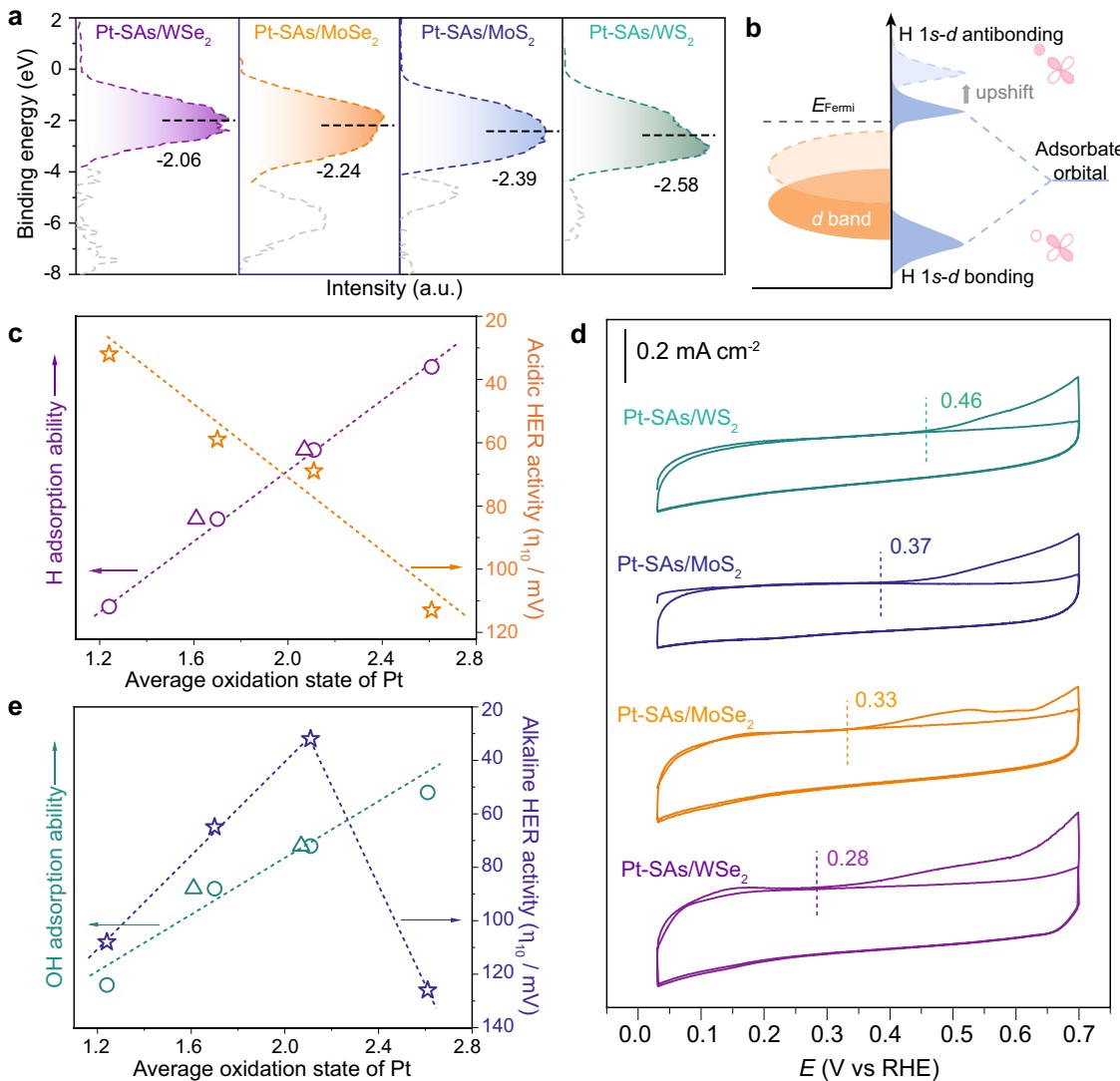

**Fig. 4 Mechanistic investigations. a** UPS valence-band spectra (VBS) of single-atom Pt relative to the Fermi level. The black dashed lines represent the position of the *d*-band centers. The gray dashed lines represent the variation in the VBS of ce-TMDs induced by single-atom Pt attachment. **b** Schematic DOS diagrams illustrating the EMSI effect on the *d*-band position of single-atom Pt, and the interaction between Pt and chemisorbed atomic hydrogen. When H is adsorbed on single-atom Pt, the interaction of the adsorbed H (H*) *s*-orbital with the Pt *d*-orbital will result in fully filled low-energy bonding states and partially filled high-energy antibonding states. **c** Relationship of average oxidation state, H adsorption ability and acidic HER activity of Pt-SAs/ TMDs. The circle and triangle represent the average oxidation state of single-atom Pt obtained from XPS and XANES, respectively. The H adsorption ability is quantified by the position of the *d*-band center. **d** CO stripping voltammetry of various single-atom Pt catalysts. The dashed lines represent the CO oxidation potentials obtained for catalysts. The scanning potential value for Pt-SAs/TMDs samples could only reach as high as 0.7 V, since higher potential resulted in the oxidation of ce-TMDs (Supplementary Fig. 36). **e** Relationship of average oxidation state, Pt–OH interaction and alkaline HER activity of Pt-SAs/TMDs. The circle and triangle are the average oxidation states of single-atom Pt obtained from XPS and XANES, respectively. The Pt–OH interaction is quantified by the CO oxidation potentials obtained from CO stripping voltammetry.

It is generally accepted that the HER process in acidic and alkaline conditions shares a similar reaction pathway, except for the generation of H* intermediates in alkaline HER through a water dissociation step (Supplementary Fig. 17). We correlated the hydrogen adsorption ability with the HER activity of four single-atom Pt catalysts in alkaline condition. We noted that with increased HBE, the alkaline HER activity of single-atom Pt catalyst increased at low oxidation states (<2.0), which was opposite to that observed in acidic HER (Supplementary Fig. 32). These results partially demonstrate that HBE of single-atom Pt— although is one of the influential factors—is not critical for the single-atom catalytic activity in alkaline HER[9]. Considering the sluggish water dissociation in the Volmer and Heyrovsky steps ($H_2O + e^- \rightarrow H^* + OH^-$ and $H^* + H_2O + e^- \rightarrow H_2 + OH^-$, *

represents an adsorption site) during alkaline HER[38], we then focused on the contribution of water dissociation.

We propose that such single-atom Pt with high oxidation state is energetically favorable for the adsorption and activation of electron-rich $H_2O$ and adsorbed OH intermediates (OH*). The deuterated effect in alkaline HER indicated that water dissociation on single-atom Pt was dominant, as evidenced by the inferior HER activities in $D_2O$ than $H_2O$ (Supplementary Fig. 33)[51,52], which was in good agreement with the Tafel slope analysis. It is well-known that the strong interaction between OH* species and surface of catalysts accelerates the water dissociation process[53]. Since OH* facilitates the removal of adsorbed CO intermediate[53], we used CO stripping voltammetry tests to measure the ability of single-atom Pt catalysts for the water dissociation (Fig. 4d).

Compared to the commercial Pt/C (0.53 V, Supplementary Fig. 34), electron-deficient single-atom Pt catalysts showed much lower onset potential of CO oxidation (0.28–0.46 V), indicating stronger Pt–OH interaction of single-atom Pt and accelerated kinetics of water dissociation (Fig. 4d). On the other hand, single-atom Pt catalyst with higher oxidation state leaded to a smaller contribution of back-donation of Pt 5d electrons to the $2\pi^*$ orbitals of CO molecule[10,54], thus weakening CO adsorption. The different CO oxidation potential also in turn reflects the variation in the electronic structure of different single-atom Pt catalysts, in line with the XPS and XANES analysis. Additionally, the d-band upshifts of the four Pt-SAs/TMDs samples (Fig. 4a and Supplementary Fig. 28) also imply an improved water dissociation ability due to the increased OH* binding energy[55–57], consistent with the CO stripping measurement (Fig. 4d). With an increase of the average oxidation state of single-atom Pt, the Pt–OH interaction became stronger while the alkaline HER activity showed a volcano-type relationship (Fig. 4e). The single-atom Pt catalyst with the average oxidation state of approximately +2 (volcano's top) exhibited exceptional alkaline HER activity.

## Discussion

In this work, we demonstrate that EMSI modulation of single-atom Pt significantly regulates the HER activity over a wide-pH range, and systematically unravel the relationship between oxidation state and HER activity of single-atom Pt. The EMSI—acting as a bridge between electronic study and catalyst design—provides a detailed explanation of the enhanced properties of supported catalysts at the electronic scale. With the length scale of catalysts shrinking to the atomic level, the EMSI effect becomes stronger and can predominate the reaction rate[24,41]. The strong EMSI between the single-atom Pt and TMDs support redistributes the electron density around the metal center with the direct formation of metal–support bonds, facilitating the electron transfer from the active metal center to the reactant. The changes in the oxidation state of single-atom Pt can be the direct effect from EMSI, which acts as a useful approach to quantitatively determine the strength of EMSI. The technical characterizations (e.g., XPS, XAS; Fig. 2) of the oxidation state pave the way for revealing the underlying mechanism of the target reaction, which in turn enhances the comprehensive understanding of EMSI and electronic structures across the length scales. Apart from changes

in the electronic structure of active sites, the stabilization effect is also a basic influence of EMSI, which suppresses the migration of single-atom metals even under operating conditions (Supplementary Figs. 23 and 24).

From the structure–activity relationship, the fine control over the oxidation state of single-atom Pt catalysts enables to achieve the optimal catalytic activity in either acidic or alkaline HER. In acidic environment, the HER performance could be well optimized through properly decreasing the oxidation state of single-atom Pt (Fig. 5a), accelerating the hydrogen desorption process. Similar to our finding, Liang's and Yao's groups have recently reported that the electron-enriched or near-zero-valence atomically dispersed Pt species are more active than the high-valence single-atom Pt for catalyzing acidic HER[29,30]. In alkaline environment, as the oxidation state of single-atom Pt increases, the HER activity also increases initially (left side of the volcano plot, Fig. 4e). Under the electrochemical condition, the charge of the metal plays a decisive role with regard to water dissociation[58,59]. Choi's group reported the counterintuitive promoting effect of CO molecule on alkaline HER of single-atom Pt[25], which could be also explained by our proposed model (Fig. 5b). After the coordination of CO (strong electron acceptor), metal-to-ligand charge transfer from Pt was increased, endowing the single-atom Pt site with higher oxidation state (ca. +2–2.3). Such single-atom Pt site was more electrophilic and favorable for the water dissociation, thus accelerating the alkaline HER. The opposite trends in acidic and alkaline HER (Fig. 4c, 4e) indicate the different mechanistic pathways of HER depending on the pH conditions, consistent with the previously reported works[25,60].

We further suggest that single-atom Pt with optimal valence state is simultaneously favorable for water dissociation, adsorption/desorption of OH* and H* (Fig. 5b, and Supplementary Fig. 35). It should be noted that with the increased oxidation state of single-atom Pt, the adsorption of H* and OH* on the catalyst were both strengthened. Although single-atom Pt with high oxidation state energetically favors the adsorption of electron-rich $H_2O$ and OH*, too strong hydrogen adsorption will also lead to the slow release of active sites and hence the sluggish HER kinetics. From a fundamental point of view, the present work reveals the atomic-level enhancement of HER thermodynamically and kinetically by optimizing the catalyst–H interaction and accelerating the water dissociation, respectively. The single-atom Pt catalyst with optimal oxidation state (~+2), showed neither

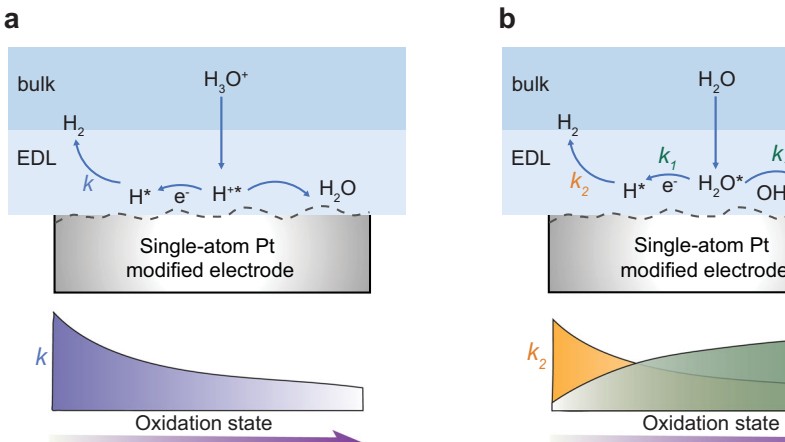

**Fig. 5 EMSI modulation mechanism of electrocatalytic HER on the single-atom Pt. a** Surface intermediate on single-atom Pt modified electrode in the acidic condition. $k$ represents the kinetics rate of $H_2$ desorption. With the increase in oxidation state of single-atom Pt, the kinetics rate of $H_2$ desorption ($k$) decreases. **b** Surface intermediate on single-atom Pt modified electrode in the alkaline condition. $k_1$ and $k_2$ represent the kinetics rates of water dissociation and $H_2$ desorption, respectively. With the increase in oxidation state of single-atom Pt, the kinetics rate of water dissociation ($k_1$) increases while the kinetics rate of $H_2$ desorption ($k_2$) decreases. EDL represents the electric double layer.

too strong Pt–H interaction to release hydrogen, nor too weak Pt–OH interaction to dissociate water, which dramatically contributes to the overall alkaline HER.

To date, three widely adopted theories have emerged to explain the alkaline HER mechanism: water dissociation theory (hydroxyl binding energy, OHBE), hydrogen binding energy (HBE) theory, or interface water and/or anion transfer theory[9]. It still remains unclear which descriptor governs the alkaline HER. At the atomic level, our results show that the two descriptors (OH* and H*) co-determined the rate of alkaline HER (Fig. 5b), which sheds light on the long-standing puzzle about HER mechanism. Although a quantitative contribution of the water dissociation and hydrogen desorption in alkaline HER is beyond the scope of the current work, the structure–activity relationship bridges a previously unconsidered link between oxidation state and wide-pH-range HER activity. Owing to the limitation of the EMSI-induced TMDs support modulation in the current work, the oxidation state of single-atom Pt was mostly restricted to a record level of ca. +1−+2.6. Further, more researches about the structure–activity relationship could be extended to the near-zero-valence single-atom metal or even negatively charged ultrasmall metal clusters. Apart from the oxidation state of single-atom metals, many complicated factors (e.g., coordination environment, reactive interface, interfacial water orientation)[44,53,58–62] could also be considered for the development of structure–activity relationship and the rational design of high-performance HER catalysts.

In summary, the EMSI-induced variation in oxidation state of single-atom Pt catalyst effectively modulates the acidic and alkaline HER. The improved HER arises from optimized thermodynamics of HER for hydrogen adsorption and accelerated reaction kinetics for water dissociation. The oxidation state of single-atom Pt controls the catalytic activity towards HER by virtue of modulating the Pt–H/Pt–OH interactions. Such atomic-level understanding of the structure–activity relationship helps to shed more insights into other single-atom electrocatalytic reduction reactions, which also involve water dissociation or hydrogen adsorption/desorption steps in carbon dioxide reductions and nitrogen reduction reactions.

## Methods

**Materials**. Molybdenum(IV) sulfide (MoS$_2$) powder, molybdenum(IV) selenide (MoSe$_2$) powder, tungsten(IV) sulfide (WS$_2$) powder, n-butyllithium in cyclohexane (2.0 M), copper(II) sulfate pentahydrate (CuSO$_4$·5H$_2$O), potassium platinum (II) chloride (K$_2$PtCl$_4$), Pt/C (20 wt%) and Nafion perfluorinated resin solution (5 wt% in a mixture of low aliphatic alcohols and water, contains 45% water) were purchased from Sigma-Aldrich (USA); tungsten(IV) selenide (WSe$_2$) powder was purchased from Aladdin Industrial Corporation (Shanghai, China). All aqueous solutions were prepared with Millipore water (resistivity of 18.2 MΩ cm).

**Synthesis of Pt-SAs/TMDs**. The single-atom Pt on different ce-TMDs nanosheets (Pt-SAs/TMDs) were synthesized through the site-specific electrodeposition method[43]. Details of the synthesis of ce-TMDs nanosheets are shown in Supplementary Methods. Briefly, a working electrode was made by drop-casting 5 μL of the well-dispersed ce-TMDs suspension (ce-MoS$_2$, ce-WS$_2$, ce-MoSe$_2$, or ce-WSe$_2$) to cover a glassy carbon electrode (GCE, 3 mm diameter). A Cu UPD process was first performed on the ce-TMDs-modified GCE by controlling the potential at +0.10 V (vs Ag/AgCl) in an Ar-saturated 0.1 M H$_2$SO$_4$ solution containing 2 mM CuSO$_4$. After the UPD process, the GCE was quickly transferred into an Ar-saturated 0.05 M H$_2$SO$_4$ solution containing 5 mM K$_2$PtCl$_4$. The electrode was kept in this solution for more than 20 min at an open circuit potential to ensure a complete galvanic replacement of Cu by Pt(II), forming atomically dispersed Pt decorated ce-TMDs nanohybrid (termed as Pt-SAs/TMDs). All the electrochemical measurements were carried out using a CHI 760E Instrument (Chenhua, China) at room temperature.

**Physical characterization**. Transmission electron microscopy (JEOL JEM-2100, Japan) and field emission electron microscopy (JEOL JEM-2800, Japan) were utilized to characterize the morphologies and elemental maps of catalysts. The atomic force microscopy (AFM) measurements were carried out using a commercial AFM (Bruker, Dimension FastScan, Icon Scanner, USA). High-angle annular dark-field-scanning transmission electron microscopy (HAADF-STEM) characterizations were carried out on a FEI Titan$^3$ G2 60-300 equipped with double aberration correctors, which was operated at 200 kV. X-ray diffraction patterns (XRD, X'TRA, Switzerland) were collected to characterize the crystal structures of samples. Inductively coupled plasma optical emission spectrometry (ICP-OES) was used to determine the loading of single-atom Pt in Pt-SAs/TMDs on a CHN-O-Rapid (German). The samples for ICP analysis were treated with aqua regia in Teflon-lined autoclaves at 230 °C for 12 h. The X-ray absorption spectroscopy at the Pt L$_3$-edge was obtained at the 14W1 beam line of the Shanghai Synchrotron Radiation Facility, using a Si(111) double-crystal monochromator operated at 3.5 GeV in fluorescence mode. The oxidation states and formal d-band hole counts of different single-atom Pt catalysts can be determined quantitatively by integrating the white-line area[44]. Specifically, the differential XANES (ΔXANES) spectra were obtained by subtracting the spectra of Pt-SAs/TMDs from that of Pt foil reference (Supplementary Fig. 12). Owing to the linear relationship between the white-line area and the oxidation states/formal d-band hole counts, these two parameters of single-atom Pt can be fitted by correlating the ΔXANES area of the references (Pt foil and PtO$_2$) and the single-atom Pt catalysts. For example, the formal d-band hole count was calculated based on the slope of 1.166 unit area per d-band hole obtained from Pt$^0$ foil (5d$^9$6s$^1$) and Pt$^{IV}$O$_2$ (5d$^6$6s$^0$) standards[29]. X-ray photoelectron spectroscopy (XPS) spectra were obtained on a SPECS Phoibos 150 (Germany) with the calibration of binding energies based on the C 1s peak energy located at 284.6 eV. The near ambient-pressure XPS (NAP-XPS) experiments were performed by using a differentially pumped electron analyzer and an in situ ambient-pressure gas cell equipped with a twin anode X-ray source (SPECS XR50, Al Kα, hγ = 1486.6 eV; Mg Kα, hγ = 1253.6 eV) under a base pressure at 0.5 mbar. The precise leak valve let gas fill the gas cell via gas line to several millibar from ultra-high vacuum in 2 min. The sample was loaded near the nozzle (300 μm) of the gas cell and illuminated to the X-ray through a 100 nm Si$_3$N$_4$ window. The high purity H$_2$ gas (99.999%, CHEM-GAS) was introduced into the cell at room temperature and liquid nitrogen trap was used to eliminant the residual water contamination during the spectra collection. All the samples were first exposed to H$_2$ for 2 h before their XPS spectra were collected. The core-level spectra of W 4f, Mo 3d, S 2p, Se 3d, and Pt 4f were measured by using the Al Kα source with the kinetic energies at around 1450 eV, 1258 eV, 1324 eV, 1432 eV, and 1414 eV, respectively. The UPS measurements were carried out by using the UVS 10/35 UV source (SPECS) in He I (21.2 eV). The UPS valence-band spectra of single-atom Pt were obtained by subtraction of the normalized ce-TMDs spectra from the Pt-SAs/TMDs spectra (Supplementary Fig. 19). The pass energy was set at 40 eV for XPS measurements and 2 eV for UPS measurements.

**Electrochemical measurements for HER**. Linear sweep voltammetry (using CHI 760E instrument, Chenhua, China) with scan rate of 20 mV s$^{-1}$ was conducted in either 0.5 M H$_2$SO$_4$ or 1.0 M KOH solution using an Ag/AgCl electrode (saturated KCl) as the reference, a graphite rod as the counter electrode, and a glassy carbon electrode (GCE) as the working electrode. The Ag/AgCl (saturated KCl) electrode was calibrated with respect to the reversible hydrogen electrode (RHE). In 1.0 M KOH solution, the loading amount of Pt-SAs/MoS$_2$, Pt-SAs/MoSe$_2$, Pt-SAs/WS$_2$, and Pt-SAs/WSe$_2$ on the GCE were 5 μg, 4.35 μg, 5.6 μg and 5 μg, respectively, E (RHE) = E(Ag/AgCl) + 1.012 V; in 0.5 M H$_2$SO$_4$ solution, the loading amount of Pt-SAs/MoS$_2$, Pt-SAs/MoSe$_2$, Pt-SAs/WS$_2$ and Pt-SAs/WSe$_2$ on the GCE were 0.625 μg, 0.55 μg, 0.715 μg and 0.65 μg, respectively, E(RHE) = E(Ag/AgCl) + 0.222 V (Supplementary Fig. 18). The commercial Pt/C catalyst ink was prepared by ultrasonically mixing 2 mg of the 12 μL 5% Nafion and 1 mL water/ethanol (v:v, 1:9) suspension for 1 h. Geometric area of GCE is 0.07065 cm$^2$. Then, 5 μL of the ink was drop-cast onto the GCE and dried naturally in air. The loading amount of Pt was about 28.3 μg$_{Pt}$ cm$^{-2}$. Before each HER LSV measurement, the catalyst on the electrode was first activated by cyclic voltammetry scanning between 0.05 V and 1.3 V (vs RHE) for 20 cycles at a scan rate of 50 mV s$^{-1}$ in Ar-saturated electrolyte (1.0 M KOH/0.5 M H$_2$SO$_4$). For CO stripping measurement, pure CO gas was first adsorbed on the working electrode at a fixed potential of 0.1 V$_{RHE}$ in a CO-saturated 1.0 M KOH electrolyte for 10 min. All the cyclic voltammograms (CVs) of CO stripping were collected after purging with Ar gas at a scan rate of 20 mV s$^{-1}$.

The reaction product of hydrogen was measured using a gas chromatograph (GC-2014, SHIMADZU) equipped with a separation column (MS-13X, 80/100 mesh, 3.2 × 2.1 mm × 2.0 m) and a thermal conductivity detector (TCD). Nitrogen was used as the carrier gas in the chromatograph. The parameters were set as follows: column temperature, 80 °C; TCD temperature, 100 °C; and bridge current, 60 mA.

The TOF of the catalysts was calculated according to the following equation:

$$TOF = I/(2F \times n) \qquad (1)$$

where I represents the measured current during linear sweep measurement, F is the Faraday constant (96,500 C mol$^{-1}$) and n is the mole amount of active Pt site. The factor 1/2 represents that two electrons are required to form one hydrogen molecule (2H$^+$ + 2e$^-$ → H$_2$).

## Data availability
All data supporting the findings in the article as well as the Supplementary Information files are available from the corresponding authors on reasonable request.

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

## Acknowledgements

The authors acknowledge the financial support from Singapore National Research Foundation under the grant of NRF2017NRF-NSFC001-007, NUS Flagship Green Energy Programme, the National Key Research and Development Program of China (2017YFA0206500), the National Natural Science Foundation of China (21902076, 21635004), and the Natural Science Foundation of the Jiangsu Province (BK20190289). The authors thank Prof. Li. Song, Prof. Shuang-Ming Chen, Dr. Yu Wang, and Wen-Jie Xu for assistance in structural analysis. The authors thank Dr. Xiao-Kun Huang and Dr. Yu Wang for his useful discussion on DFT simulations. The authors thank Dr. Zhang-Liu Tian, Dr. Da-Feng Yan, Dr. Wen-Rui Dai, Ms. Yu-Min Da, and Ms. Meng Wang for assistance in the research work. The authors also gratefully thank the Shanghai Synchrotron Radiation Facility (14W1, SSRF).

## Author contributions

Y.S., X.X. and W.C. conceived the project. Y.S., Z.M. and Y.X. carried out the synthesis of the catalysts, physical and electrochemical characterizations. Y.Y., W.H. and G.S. assisted in the data analysis. Z.H. and Y.-Y.S. performed the DFT calculations. Y.Z., F.M. and R.H. carried out and assisted in the morphology characterization. Y.S., X.X. and W.C. wrote the manuscript. All authors commented on the manuscript.

## Competing interests

The authors declare no competing interests.
