## [Peer Review File · Nature Communications]

REVIEWER COMMENTS

Reviewer #1 (Remarks to the Author):

In this manuscript, the authors reported four kinds of single-atom Pt supported on different transition metal dichalcogenides (MoS₂, WS₂, MoSe₂, and WSe₂) as efficient electrocatalysts for HER. Fundamental morphology, structure, catalytic performance and stability of the material in both alkaline and acidic media were studied. Detailed spectroscopic and electrochemical characterizations showed that the fine tailoring of the oxidation states of single-atom Pt catalysts through EMSI significantly could modulate the catalytic activities in either acidic or alkaline HER. The authors also conducted different experimental measurements to study the HER intermediates (H* and OH*), and revealed the structure-activity relationship by UPS and DFT calculations. After reading this manuscript, I suggest this manuscript should be resubmitted after addressing the following issues.

1. In the Introduction, the author should give explanation about why TMD was chosen as the support of single atoms. Besides, it is well known that transition metal dichalcogenides as semiconductors possess novel band structure. Their conductivity can be effectively tuned by heteroatom doping. In this study, the incorporation of Pt atom into MoS₂, MoSe₂, WS₂ and WSe₂ substrates may induce significant change in the conductivity, which plays a key role in the electrocatalytic process. This study should not attribute the catalytic performance enhancement to the modulation of platinum before confirming the Pt incorporation influence on the dichalcogenides substrates.
2. The quality of atomic resolution HAADF-STEM images is poor. In Fig.1 c and d, it seems to be cluster, not well-dispersed single atoms. If possible, please provide a clear picture or explain the phenomenon of cluster.
3. XPS characterization of TMDs should be supplemented. And the comparison of XPS between TMDs and Pt-SAs/TMDs should be provided. The authors should explain the electronic interaction between single atom and support.
4. The XPS characterization of Pt-SAs/TMDs after HER measurements should also be performed to determine whether the valence state of Pt is changed during the HER measurement.
5. On Page 6 line 141 and 143, the authors claimed that the oxidation states of single-atom Pt in Pt-SAs/MoS₂ and Pt-SAs/MoSe₂ samples were 1.61 and 2.07; the number of d-band hole for Pt-SAs/MoSe₂ and Pt-SAs/MoS₂ was estimated to be 2.590, and 2.205. The specific calculation method of oxidation states and the number of d-band holes should be given.
6. In Supplementary Note 1. Calculation of mass activity for HER, the calculation of mass activity for Pt/C is missing, but its corresponding values appear in Fig 3b and Fig 3f. Besides, the tested HER activity of Pt/C in acid and alkaline medium is worse than the reported activity in many literature. If possible, please retest to determine whether the value is accurate.

7. The substrates used in this study is active to HER, how the authors ruled out their influence on the result?

8. In line 258~266, the authors determine the water dissociation ability by “CO stripping voltammetry”. However, this is not convincing. The authors mention that the water dissociation can be strengthened by the strong interaction between OH* and catalysts surface. And the OH* can facilitate the removal of adsorbed CO intermediate. Therefore, one can determine the water dissociation capability by checking the CO removal ability. It is noteworthy that the removal of CO intermediate is not only affected by the OH* interaction (e.g. the substrate). Hence, the measurement of water dissociation ability in the manuscript is not persuasive.

9. To compare the HER performance of Pt-SAs/TMDs in this work, the authors should cite several relative references based on Pt catalyst in Table 4 and 5.

Reviewer #2 (Remarks to the Author):

In this work, the author reported that the single-atom Pt catalysts are located on TMDs supports (MoS₂, WS₂, MoSe₂, and WSe₂). The oxidation state of monoatomic Pt controls the catalytic activity of HER by adjusting the Pt–H/Pt–OH interaction. There is a very large amount of experimental data. Thus, I think this manuscript meet the high criteria of Nature Communications and recommend its publication after mini revision. The following comments should be taken for more consideration.

1. It would be better if the TEM of the catalysts were provided after 1000 cycles stability tests.

2. The format for the references is not uniform, please check.

3. In this work, the authors tune the electronic metal–support interaction by change the support, for example, MoS₂, WS₂, MoSe₂, and WSe₂. If the authors try to tune the electronic metal–support interaction (such as the local coordination environment of single-atom) on the same support? Some related works about this issue can be refereed, for example, Modulating the local coordination environment of single-atom catalysts for enhanced catalytic performance, *Nano Res.*, 2020, 13, 1842–1855; Atomic-Level Modulation of Electronic Density at Cobalt Single-Atom Sites Derived from Metal–Organic Frameworks: Enhanced Oxygen Reduction Performance, *Angew. Chem. Int. Ed.*, 2020, DOI: 10.1002/anie.202012798; etc.

4. More discussion is needed to emphasize the advantages of the EMSI effect in the structure and activity relationship section.

Reviewer #3 (Remarks to the Author):

In this paper, the authors investigated the catalytic properties of single-atom catalysts based on Pt

atoms at TMDs, and found very promising performance regarding HER. The study combines state-of-the-art methods of the synthesis and the characterization of the catalysts, electrochemical HER study and the discussion on the structure and the reactivity relationship.

Although the studied systems and results are very interesting and the methodology combined approaches able to illuminate a microscopic picture of investigated catalysts, unfortunately, I can not recommend the publication of the paper due to significant similarity to the Ref. 31 - "Site-specific electrodeposition enables self-terminating growth of atomically dispersed metal catalysts" - recently published work of several co-authors of the present manuscript (Nature Communication 11, 4558 (2020)).

In Ref. 31, the authors mainly studied Pt atoms at MoS₂ but also included results for Pt at WS₂ and MoSe₂, as well as a few other metals at MoS₂. In this paper, they focused on Pt-based catalysts. In addition to three substrates considered in Ref. 31, WSe₂ was included as the 4th support.

In the present paper the main results, highlighted in the abstract, are increased activity of Pt₁/WS₂ and Pt₁/MoSe₂, compared to Pt/C and the correlation of the activity and the Pt interaction with the support. However, the activity of several of the inspected catalysts was already compared in Fig. 24, Supported Information (SI) of Ref. 31. In Table 4, SI of Ref. 31 were also presented DFT results for the binding energies of Pt adatom at MoS₂, WS₂ and MoSe₂ (these results are partially presented again in Table 3, the SI of the present manuscript).

Due to apparent similarity in studied systems, methodology, and results, the present manuscript could be considered as a follow up to Ref. 31, and thus lacks sufficient novelty compulsory for the publication in Nature Communications.

The present manuscript, which I consider as a nice work, could be improved by including a better description of the Pt interaction with TMDs. The effect of the support should be presented in more detail. These details can be easily obtained from DFT calculations, already performed by the authors:

1. In Fig. 4a are shown positions of d-bands from UPS valence-band spectra. The authors should include DOS projected on Pt-d orbitals and the d-states occupation numbers, calculated using DFT.
2. In Table 3 of the SI are listed Pt binding energies. Even more useful information would be the energy barriers for Pt diffusion between the nearest adsorption sites. The calculated activation energies will demonstrate the structural stability of the studied single-atom catalysts. This is an important issue, studied by Li et al. (ACS Nano 11, 3392 (2017)), where they claimed that Pt atoms at the ideal 2H-MoS₂ were very mobile in the room temperature and only those trapped at the S-vacancies were structurally stable. In the present study, one of the supports was 1T phase of MoS₂, so the picture might be different from 2H-MoS₂, but the authors should clarify this by carrying out the corresponding DFT calculations.

Responses to Reviewers

Reviewer #1:

In this manuscript, the authors reported four kinds of single-atom Pt supported on different transition metal dichalcogenides (MoS₂, WS₂, MoSe₂, and WSe₂) as efficient electrocatalysts for HER. Fundamental morphology, structure, catalytic performance and stability of the material in both alkaline and acidic media were studied. Detailed spectroscopic and electrochemical characterizations showed that the fine tailoring of the oxidation states of single-atom Pt catalysts through EMSI significantly could modulate the catalytic activities in either acidic or alkaline HER. The authors also conducted different experimental measurements to study the HER intermediates (H* and OH*), and revealed the structure-activity relationship by UPS and DFT calculations. After reading this manuscript, I suggest this manuscript should be resubmitted after addressing the following issues.

Response: Thank you for your summary. We appreciate your efforts in reviewing our manuscript. We have revised the manuscript accordingly. Our point-by-point responses are detailed below.

1. In the Introduction, the author should give explanation about why TMD was chosen as the support of single atoms. Besides, it is well known that transition metal dichalcogenides as semiconductors possess novel band structure. Their conductivity can be effectively tuned by heteroatom doping. In this study, the incorporation of Pt atom into MoS₂, MoSe₂, WS₂ and WSe₂ substrates may induce significant change in the conductivity, which plays a key role in the electrocatalytic process. This study should not attribute the catalytic performance enhancement to the modulation of platinum before confirming the Pt incorporation influence on the dichalcogenides substrates.

Response: We thank this reviewer for the valuable comments and concern. To highlight our design concept and make the current work more precise, we have made the following revisions and discussion:

(1) explanation of why TMD was chosen as the support of single atoms in the Introduction (paragraph 4, Fig. 1a):

“Transition metal dichalcogenides (TMDs) have been widely used as the supports for immobilizing SAMCs in heterogeneous catalysis [*Chem. Rev.* 2019, 119, 1806; *Chem. Rev.* 2020, 120, 11810; *Chem* 2020, 6, 885; *Energy Environ. Sci.*, 2015, 8, 1594; *Nat. Chem.* 2017, 9, 810; *Nat. Commun.*, 2019, 10, 5231]. Compared with SAMCs supported on carbon-based materials [*Angew. Chem. Int. Ed.*, 2020, DOI: 10.1002/anie.202012798; *J. Am. Chem. Soc.* 2020, 142, 8431; *J. Am. Chem. Soc.* 2019, 141, 20118], the electronic structure of single-atom metals supported on TMDs is usually adjusted by both the anchoring atom and the neighboring transition metal atoms with relatively high atomic number, which affords a more flexible and complex coordination environment to regulate the catalytic activity [*Nano Res.*, 2020, 13, 1842; *Adv. Mater.* 2020, 32, 2003300]. Owing to the various well-defined band structures of TMDs (*e.g.* MoS₂, WS₂, MoSe₂, and WSe₂, Fig. 1a), the core anchoring chalcogen (S, Se) and the neighboring transition metal (Mo, W) can synergistically regulate the electronic structure of SAMC through EMSI. The tuneable *d*-orbital state of single-atom Pt changes the adsorption energy of reactants on metal atoms and thus influences the catalytic activity of HER (Fig. 1a).”

Fig. 1 | Rational design and construction of single-atom Pt catalysts. (a) Electronic metal–support interactions (EMSI) modulation of single-atom Pt for catalyzing HER. Left: schematic structure of single-atom Pt on TMDs material. The grey, purple, and green spheres represent the chalcogen (sulphur/selenium), transition metal (molybdenum/tungsten), and platinum, respectively. The electronic structure of single-atom Pt was modulated by two-dimensional TMDs through charge delocalization, enabling the single-atom Pt to take slightly positive charge (Pt^{δ+}). The structural unit of single-atom Pt was circled by the orange dashed line and further enlarged above. Top right: schematic diagram of the band edges of TMDs. The conduction band minimum

(CBM)/valence band maximum (VBM) band edges of TMDs (theoretical values) refer to ref. 42 [*Chem. Soc. Rev.* 2018, 47, 6845]. The schematic band structure—showing the electron affinity and ionization potential of various TMDs—provides a guideline for rationalizing the EMSI modulation of single-atom Pt. Bottom right: schematic illustrating that the *d*-state shift of single-atom Pt induced by EMSI regulates the catalytic performance of HER.

(2) We agree with the reviewer that heteroatom doping effectively increases the conductivity of substrate and further enhances the catalytic activity [*e.g.* *J. Am. Chem. Soc.* 2017, 139, 15479; *Adv. Mater.* 2013, 25, 5807; *Chem. Sci.* 2011, 2, 1262; *Energy Environ. Sci.* 2015, 8, 1594]. However, the structure in our work is protrusion-shaped single-atom Pt straddled atop Mo/W, which is different from the single-atom Pt incorporation into TMD lattice [for example, *Energy Environ. Sci.* 2015, 8, 1594] (detailed discussion and comparison shown below). The catalytically active Pt is adsorbed atop TMDs and can expose more unsaturated active sites to reactive species, which reduces the influence of TMDs support for HER enhancement.

We further conducted the thiocyanate ion (SCN^-) poison experiment (Supplementary Fig. 26), which suggested that the HER activity dominantly derives from the single-atom Pt and thus the catalytic performance enhancement is mainly attributed to the EMSI modulation of Pt.

Following this valuable suggestion, we have added the relevant descriptions, figures and tables in the revised manuscript and supplementary materials (supplementary Note 3, Supplementary Fig. 26, and Supplementary Tab. 8). The details are shown below:

“We further verified the active sites for HER and the negligible contribution of TMDs support to HER. The thiocyanate ions (SCN^-) poison experiment of the Pt-SAs/TMDs samples was conducted to efficiently block the Pt sites for acidic HER [*Nat. Commun.*

10, 5231 (2019); *Nat. Commun.* 11, 4558 (2020)]. Upon the addition of SCN^- , the HER current of all the Pt-SAs/TMDs samples decreased dramatically approaching near zero, confirming that HER activity dominantly derives from the single-atom Pt and the catalytic performance enhancement is mainly attributed to the EMSI modulation of Pt (Supplementary Fig. 26). This phenomenon is distinct from the Pt-doping case, where Pt atoms are incorporated into the TMD lattice and chalcogen atoms are reported as the active sites of HER (Supplementary Note 3, and Supplementary Tab. 8) [*Energy Environ. Sci.* 2015, 8, 1594]. Additionally, the Tafel behavior of Pt-SAs/TMDs in acidic HER ($\sim 30 \text{ mV dec}^{-1}$) resembles that of the commercial Pt, indicating that the catalytic reaction on single-atom Pt contributed mostly to the HER. On the contrary, it has been reported that Pt-doped MoS_2 showed a Tafel slope of 96 mV dec^{-1} , close to that of pure MoS_2 ($\sim 100 \text{ mV dec}^{-1}$) [*Energy Environ. Sci.* 2015, 8, 1594].”

Supplementary Figure 26. Current–time curves of the Pt-SAs/TMDs samples before and after the addition of 10 mM thiocyanate ions (SCN^-) at -0.28 V (vs Ag/AgCl) in $0.5 \text{ M H}_2\text{SO}_4$.

Supplementary Note 3. Comparison of single-atom Pt adsorbed atop MoS₂ and Pt-doped MoS₂ cases

In order to elucidate the advantage of single-atom Pt supported atop TMDs (Pt-SAs/TMDs in this work) for the atomic-level electronic modulation for HER enhancement, we further compared the adsorbed single-atom Pt with the previously reported Pt-doped MoS₂ material [*Energy Environ. Sci.*, 2015, 8, 1594]. In our system, we can see that single-atom Pt adsorbed atop MoS₂ (Pt-SAs/MoS₂) was dramatically poisoned by SCN⁻ owing to the blocking of active Pt sites, resulting in a large decrease of HER current approaching zero (Supplementary Fig. 26). Similarly, Bao *et al.* also found that the activity of Pt adsorbed on MoS₂ drops quickly after adding methanol because the exposed Pt atoms atop MoS₂ support were easily poisoned by methanol (Fig. R1). On the contrary, Pt-doped MoS₂ showed negligible current decrease and excellent poisoning resistance ability with the introduction of methanol (Fig. R1). Through DFT calculations, owing to the steric hindrance, the H atom cannot adsorb on the doped Pt atom, while the preferred adsorption site is the neighboring unsaturated S atom with ΔG_H of ~ 0 eV (Supplementary Tab. 8). In the adsorbed Pt case, the preferred adsorption site for H atom is the Pt atom with ΔG_H of also ~ 0 eV, while the neighboring S atom shows very large ΔG_H value of ~ 1.94 eV (Supplementary Tab. 8). In this case, they came to the conclusion that the active sites of HER in such Pt-doped MoS₂ should originate from the S atoms rather than the Pt atoms. Compared with the Pt-doping case, the single-atom Pt adsorbed atop MoS₂ in our work can expose more unsaturated active sites to reactive species, which reduces the influence of ce-TMDs support for HER enhancement.

Figure R1. Chronoamperometric responses on pure MoS₂ (FL-MoS₂), Pt-doped MoS₂ (Pt-MoS₂) and MoS₂-supported Pt (Pt/MoS₂) in a O₂-saturated solution of 1 M NaOH. A 10% (v/v) methanol was injected into the electrolyte solution at 200 s (adopted from *Energy Environ. Sci.*, 2015, 8, 1594, Fig. S8). Pt-doped MoS₂ showed excellent poisoning resistance ability with the introduction of methanol, while MoS₂-supported Pt are easily poisoned by methanol as revealed by the significant decrease in current.

Supplementary Table 8. Comparison between the previously reported Pt-doped MoS₂ and the single-atom Pt adsorbed atop MoS₂ for electrocatalytic HER.

Catalyst	Structure	Scheme	Active site	Tafel slope (mV dec ⁻¹)	ΔG_H	Reference
Pt-SAs/MoS ₂	Pt adsorbed atop MoS ₂		Pt atoms	31		This work
Pt-doped MoS ₂	Single-atom Pt doping (Mo substitution)		S atoms	96		Energy Environ. Sci. , 2015, 8, 1594

Additionally, unlike 2H-MoS₂ where the catalytic activity arises from the edges, experimental and theoretical results from Chhowalla *et al.* and Jiang *et al.* suggested that the active sites of the chemically exfoliated 1T MoS₂ nanosheets are mainly the S atoms located on the basal plane and the contribution of the metallic edges to the overall HER efficiency is relatively small [*Nano Lett.* 2013, 13, 6222; *ACS Catal.* 2016, 6, 4953]. Thus, the Pt attachment on chalcogen atoms in the basal plane of 1T-TMDs can largely reduce the influence of TMDs supports for electrocatalytic HER, owing to the occupation of active sites of 1T-TMDs and the steric hindrance.

2. The quality of atomic resolution HAADF-STEM images is poor. In Fig.1 c and d, it seems to be cluster, not well-dispersed single atoms. If possible, please provide a clear picture or explain the phenomenon of cluster.

Response: We thank the reviewer for pointing this out. We are sorry about the imperfect characterization of some SAMCs in the original text. We have very carefully re-characterized those SAMCs with freshly prepared samples to confirm the atomic dispersion of these samples (see updated data in Fig.1 d and e, which is Fig.1 c and d in the original text).

Figure 1. Atomic-resolution HAADF-STEM images for (c) Pt-SAs/MoS₂, (d) Pt-SAs/MoSe₂, (e) Pt-SAs/WS₂, (f) Pt-SAs/WSe₂ (scale bars: 5 nm) and the corresponding elemental mappings (right side, scale bars: 100 nm).

3. XPS characterization of TMDs should be supplemented. And the comparison of XPS between TMDs and Pt-SAs/TMDs should be provided. The authors should explain the electronic interaction between single atom and support.

Response: We thank the reviewer for this valuable advice. XPS characterizations of TMDs before and after the decoration of single-atom Pt have been supplemented in the supplementary material (Supplementary Fig. 11).

Relevant description has been added in the highlighted manuscript: “The binding energies of Mo 3d/W 4f and S 2p/Se 3d in the supports decreased slightly (Supplementary Fig. 11) while the binding energy of Pt 4f increased, indicating that electrons were transferred from Pt to ce-TMDs supports”.

Supplementary Figure 11. Mo 3d (a) and S 2p (b) XPS spectra of Pt-SAs/MoS₂ and ce-MoS₂. Mo 3d (c) and Se 3d (d) XPS spectra of Pt-SAs/MoSe₂ and ce-MoSe₂. W 4f (e) and S 2p (f) XPS spectra of Pt-SAs/WS₂ and ce-WS₂. W 4f (g) and Se 3d (h) XPS spectra of Pt-SAs/WSe₂ and ce-WSe₂.

4. The XPS characterization of Pt-SAs/TMDs after HER measurements should also be performed to determine whether the valence state of Pt is changed during the HER measurement.

Response: We thank the reviewer for this valuable suggestion. XPS characterizations of Pt-SAs/TMDs after HER measurements revealed the unchanged valence state (within the accuracy of our instrument) of single-atom Pt during HER (Tab. R1).

Relevant figures and description has been added in the revised manuscript and supplementary material (Supplementary Fig. 24, and Supplementary Tab. 5).

Table R1. Comparison of average oxidation state of the single-atom Pt before and after HER measurements.

Catalyst	Pt-SAs/WS ₂	Pt-SAs/MoS ₂	Pt-SAs/MoSe ₂	Pt-SAs/WSe ₂
Before HER	1.24	1.71	2.11	2.61
After HER	1.23	1.74	2.13	2.60

Supplementary Figure 24. Pt 4f XPS spectra of the Pt-SAs/TMDs samples before and after the HER measurements. Quantitative peak deconvolution and integration of XPS analysis showed that after the HER measurements, the average oxidation states of Pt in Pt-SAs/WS₂, Pt-SAs/MoS₂, Pt-SAs/MoSe₂, and Pt-SAs/WSe₂ were 1.23, 1.74, 2.13, and 2.60, respectively (Supplementary Tab. 5).

Supplementary Table 5. Average oxidation state of Pt calculated from the XPS spectra of the Pt-SAs/TMDs catalysts after HER measurements.

Sample	Binding energy of Pt 4f _{7/2} (eV)	Content (%)	Average oxidation state of Pt after HER measurement
Pt-SAs/WS ₂	72.30 (Pt ^{II})	61.3	1.23
	71.10 (Pt ⁰)	38.7	
Pt-SAs/MoS ₂	73.50 (Pt ^{IV})	8.2	1.74
	72.20 (Pt ^{II})	70.6	
	71.50 (Pt ⁰)	21.2	
Pt-SAs/MoSe ₂	73.78 (Pt ^{IV})	6.7	2.13
	72.37 (Pt ^{II})	93.3	
Pt-SAs/WSe ₂	73.95 (Pt ^{IV})	30.2	2.60
	72.62 (Pt ^{II})	69.8	

5. On Page 6 line 141 and 143, the authors claimed that the oxidation states of single-atom Pt in Pt-SAs/MoS₂ and Pt-SAs/MoSe₂ samples were 1.61 and 2.07; the number of d-band hole for Pt-SAs/MoSe₂ and Pt-SAs/MoS₂ was estimated to be 2.590, and 2.205. The specific calculation method of oxidation states and the number of d-band holes should be given.

Response: We thank the reviewer for this valuable suggestion. The specific calculation method of oxidation states and the number of *d*-band holes have been provided in the revised manuscript (Methods, Physical characterization):

“The oxidation states and formal *d*-band hole counts of different single-atom Pt catalysts can be determined quantitatively by integrating the white-line area [*Nat. Commun.* 2019, 10, 4500]. Specifically, the differential XANES (Δ XANES) spectra were achieved by subtracting the spectra of Pt-SAs/TMDs from that of Pt foil reference (Supplementary Fig. 12). Owing to the linear relationship between the white-line area

and the oxidation states/formal d -band hole counts, these two parameters of single-atom Pt can be fitted by correlating the Δ XANES area of the references (Pt foil and PtO₂) and the single-atom Pt catalysts. For example, the formal d -band hole count was calculated based on the slope of 1.166 unit area per d -band hole obtained from Pt⁰ foil (5d⁹6s¹) and Pt^{IV}O₂ (5d⁶6s⁰) standards [*Nat. Commun.* 2020, 11, 1029].”

Supplementary Figure 12. Normalized Δ XANES spectra for Pt L₃-edge using Pt foil as the reference. The oxidation states and the d -band hole counts were fitted by integrating the white-line area from 11560 to 11580 eV.

6. In Supplementary Note 1. Calculation of mass activity for HER, the calculation of mass activity for Pt/C is missing, but its corresponding values appear in Fig 3b and Fig 3f. Besides, the tested HER activity of Pt/C in acid and alkaline medium is worse than the reported activity in many literature. If possible, please retest to determine whether the value is accurate.

Response: We thank the reviewer for this valuable advice. We have carefully retested the HER activity of commercial Pt/C in either alkaline or acidic conditions (Method,

updated data shown in Supplementary Figs. 20 and 25), and also provided the calculation of mass activity for Pt/C in Supplementary Note 1 and Fig. 3e and f.

(1) Before each HER measurement, the Ag/AgCl (saturated KCl) electrode was calibrated with respect to reversible hydrogen electrode (RHE) in either 1.0 M KOH or 0.5 M H₂SO₄, thus making the results of subsequent tests more convincing:

Figure R2. Calibration of the Ag/AgCl reference electrode in 0.5 M H₂SO₄ (a) and 1.0 M KOH (b) solutions for the HER tests of commercial Pt/C, respectively.

(2) The commercial Pt/C catalyst ink was prepared by ultrasonically mixing 2 mg of the 12 μ L 5% Nafion and 1 mL water/ethanol (v:v, 1:9) suspension for 1 h. Geometric area of GCE is 0.07065 cm². Then, 5 μ L of the ink was drop-cast onto the GCE and dried naturally in air. The loading amount of Pt was about 28.3 μ g_{Pt} cm⁻². Before each HER LSV measurement, the catalyst on the electrode was first activated by cyclic voltammetry scanning between 0.05 V and 1.3 V (vs RHE) for 20 cycles at a scan rate of 50 mV s⁻¹ in Ar-saturated electrolyte (1.0 M KOH/0.5 M H₂SO₄).

Six independent Pt/C-modified glassy carbon electrodes were tested in Ar-saturated acidic (0.5 M H₂SO₄) and alkaline (1.0 M KOH) media at a scan rate of 20 mV s⁻¹ (at 25 °C), respectively. The final plots (Supplementary Figs. 20 and 25) shown in the paper were the average data obtained from the six independent tests (Fig. R3).

Figure R3. Six independent HER tests of commercial Pt/C in 0.5 M H₂SO₄ (a) and 1.0 M KOH (b) solutions at a scan rate of 20 mV s⁻¹.

Supplementary Figure 20. (a, b) HER polarization curves in 1.0 M KOH for Pt-SAs/MoSe₂ and commercial Pt/C at a scan rate of 20 mV s⁻¹, and their corresponding Tafel plots derived from the early stage of HER LSV curves.

Supplementary Figure 25. (a, b) HER polarization curves in 0.5 M H₂SO₄ for Pt-SAs/WS₂ and commercial Pt/C at a scan rate of 20 mV s⁻¹, and their corresponding Tafel plots derived from the early stage of HER LSV curves.

(3) Mass activity calculation of commercial Pt/C (updated in Supplementary Note 3, and Fig. 3b and f):

$$\frac{13.25 \times 0.07065 \times 10^{-3}}{2 \times 5 \times 10^{-3} \times 20\%} = 0.468 \text{ A mg}^{-1} \text{ (Alkaline HER)}$$

$$\frac{85.6 \times 0.07065 \times 10^{-3}}{2 \times 5 \times 10^{-3} \times 20\%} = 3.02 \text{ A mg}^{-1} \text{ (Acidic HER)}$$

Fig. 3 (b) Alkaline and (f) acidic HER comparison of overpotentials required to achieve a current density of 10 mA cm⁻² (black arrow, left axis) and mass activities (normalized by the Pt loading, red arrow, right axis) at -100 mV *versus* RHE for various Pt-SAs/TMDs samples.

(4) To further confirm that our data are within a reasonable range for HER activity, we also listed the previously reported HER activity in many literatures for double-check:

Table R1. Comparison of the alkaline HER performance for commercial Pt/C catalysts described in literature.

Loading ($\mu\text{g}_{\text{Pt}} \text{ cm}^{-2}_{\text{disk}}$)	Current density (mA cm ⁻²)	η_{10} (mV)	Tafel slope (mV dec ⁻¹)	Reference
--	---	---------------------	--	-----------

28.3	8.5@70 mV	82.5	57	This work
5.1	~8@100 mV	108.9	78.8	J. Am. Chem. Soc. 2018, 140, 9046
14.28	13.52@70 mV	55.6	72.7	Angew. Chem. Int. Ed., 2019, 58, 19060
14.8	8.7@70 mV	77	46	ACS Catal. 2019, 9, 10870
15.31	14.9@70 mV	55	108.7	Angew. Chem. Int. Ed. 2018, 57, 11678
15.3	3.3@70 mV	>100	NA	Angew. Chem. 2016, 128, 13051
12.76	~5@70 mV	94	48.6	Adv. Funct. Mater. 2020, 30, 2004310
17	~12@70 mV	62	44	Adv. Mater. 2018, 30, 1801741
8	3.42@70 mV	>100	NA	Nat. Commun., 2017, 8, 14580

Table R2. Comparison of the acidic HER performance for commercial Pt/C catalysts described in literature.

Loading ($\mu\text{g}_{\text{Pt}} \text{ cm}_{\text{disk}}^{-2}$)	Current density (mA cm^{-2})	η_{10} (mV)	Tafel slope (mV dec^{-1})	Reference
28.3	51.2@70 mV	33.7	29	This work
640	NA	>50	32	Energy Environ. Sci., 2015, 8, 1594
27	~28@100 mV	~55	30	Nat. Commun., 2013, 4, 1444
56.6	~68@75 mV	42.03	31	ACS Catal. 2019, 9, 8213
102	~55@80 mV	~32	35	Nat. Energ. 2019, 4, 512
20.38	NA	26.3	27.1	J. Am. Chem. Soc. 2020, 142, 17250

NA	~22@100 mV	>35	31	Energ. Environ. Sci. 2017, 10, 2450
56.6	13.87@50 mV	45.2	31	Nano Energy 2019, 63, 103849
17	~40@70 mV	39	29	Int. J. Hydrog. Energy 2016, 42, 18193.
51	~17@50 mV	34	46	J. Mater. Chem. A, 2019, 7, 6285
NA	NA	43	31	J. Colloid Interface Sci. 2017, 505, 14
22.65	~38@50 mV	~35	30	Nanoscale, 2017, 9, 10138
25	~40@50 mV	31	31	Nanoscale, 2020, 12, 20456
26.5	~60@100 mV	36	32	ACS Appl. Mater. Interfaces 2020, 12, 9600
13	NA	~30	30	ACS. Appl. Mater. Interface 2017, 9, 3596
15.2	~70@100 mV	42	36	Carbon 2019, 146, 116
NA	~25@50 mV	41	29	J. Electroanal. Chem. 2018, 822, 10
51	~50@50 mV	45.1	28.7	J. Mater. Chem. A 2019, 7, 20239

7. The substrates used in this study is active to HER, how the authors ruled out their influence on the result?

Response: We agree with the reviewer that the two-dimensional transition metal dichalcogenides (MoS₂, WS₂, MoSe₂, and WSe₂) are also active to HER. However, in either acidic or alkaline HER, all the pure ce-TMDs supports showed negligible electrocatalytic HER activity compared with those of Pt-SAs/TMDs, especially in the low overpotential region (-0.3–0 V vs RHE) (Supplementary Fig. 19). The thiocyanate ions (SCN⁻) poison experiment further confirmed that the HER activity dominantly

derives from the single-atom Pt and the catalytic performance enhancement is mainly attributed to the EMSI modulation of Pt (see Question 1 or Supplementary Fig. 26). Detailed discussion has also been provided (see Question 1 or Supplementary Note 3) to compare the single-atom Pt adsorbed atop TMDs with the Pt-doped TMDs case. The active sites of our system are Pt atoms whereas the active sites of Pt-doped case are S atoms [*Energy Environ. Sci.*, 2015, 8, 1594]. All these results indicate that single-atom Pt plays a central role in the HER, implying that TMDs have less effect on the construction of the atomic-level structure-activity relationship.

We further correlated the loading amount of single-atom Pt (for example, Pt-SAs/MoS₂) with the HER activity. A nearly linear correlation (Fig. R4) corroborated that single-atom Pt contributed mostly to the HER, which rationalizes the EMSI modulation of our single-atom system for the mechanistic investigation of structure-activity relationship.

Figure R4. Linear plot of HER activity (@-100 mV vs RHE) versus loading amount of single-atom Pt of Pt-SAs/MoS₂ materials.

8. In line 258~266, the authors determine the water dissociation ability by “CO stripping voltammetry”. However, this is not convincing. The authors mention that the water dissociation can be strengthened by the strong interaction between OH* and catalysts surface. And the OH* can facilitate the removal of adsorbed CO intermediate. Therefore, one can determine the water dissociation capability by checking the CO

removal ability. It is noteworthy that the removal of CO intermediate is not only affected by the OH* interaction (e.g. the substrate). Hence, the measurement of water dissociation ability in the manuscript is not persuasive.

Response: We thank the reviewer for this comments. There has been consensus that the onset potential of CO stripping provides a sensitive measure of the lowest potential for OH_{ad} present on the surface, and thus the binding energy of OH_{ad} [*Nat. Commun.* 2019, 10, 4876; *Angew. Chem.* 2017, 129, 15800; *Sci. Adv.* 2016, 2, e1501602; *Nat. Mater.* 2012, 11, 550; *Nat. Chem.* 2013, 5, 300; *J. Am. Chem. Soc.* 2007, 129, 11033; *Energy Environ. Sci.* 2015,8, 177; *ChemSusChem* 2018, 11, 2388; *J. Mater. Chem. A* 2019, 7, 13635]. The formation of OH_{ad} can enhance the oxidation of CO_{ad}, and the CO_{ad} oxidation peak shifts to more negative potentials, which indicates the easier formation of surface-bound oxygen species [*J. Phys. Chem.* 1994, 98, 617; *Langmuir* 2000, 16, 522]. The CO oxidation on Pt usually follows a Langmuir–Hinshelwood surface reaction between the adsorbed CO and oxygen-containing species [*J. Phys. Chem.* 1964, 68, 70]. To date, the CO stripping experiment is the most reliable tool and convincing characterization for monitoring the generation of reactive hydroxyl species, since the anodic current of CO oxidation has to be triggered by the reactive hydroxyl species [*J. Am. Chem. Soc.*, 2007, 129, 11033; *Energy Environ. Sci.*, 2015,8, 177]. This method has been widely used in many research works for measuring the binding energy of OH_{ad} [e.g., *Nat. Commun.* 2019, 10, 4876; *Angew. Chem.* 2017, 129, 15800; *Sci. Adv.* 2016, 2, e1501602; *Nat. Mater.* 2012, 11, 550; *Nat. Chem.* 2013, 5, 300; *J. Am. Chem. Soc.* 2007, 129, 11033; *Energy Environ. Sci.* 2015,8, 177; *ChemSusChem* 2018, 11, 2388; *J. Mater. Chem. A* 2019, 7, 13635]. According to the Brønsted-Evans-Polanyi principle, the ability of a catalyst to dissociate water is correlated with the OH adsorption energy [*Science* 2011, 334, 1256; *Angew. Chem. Int. Ed.* 2020, 59, 10934].

Additionally, due to the EMSI modulation, various density of states patterns were obtained from the DFT calculation for the Pt-*d* orbitals of the four Pt-SAs/TMDs (Supplementary Fig. 28). The *d*-band upshifts of the four Pt-SAs/TMDs samples (Fig.

4a and Supplementary Fig. 28) also imply an improved water dissociation ability due to the increased OH_{ad} binding energy [*Angew. Chem. Int. Ed.* 2019, 58,19060; *Angew. Chem. Int. Ed.* 2020, 59, 10934; *Phys. Chem. Chem. Phys.* 2010, 12, 5694], consistent with the CO stripping measurement. Relevant figures and description have been added in the revised manuscript and supplementary material (Supplementary Fig. 28).

Supplementary Figure 28. Density of states projected on Pt-*d* orbitals of the Pt-SAs/TMDs. The white dashed lines represent the *d*-band center calculated by DFT.

CV measurements under non-catalytic conditions were usually used to measure the Pt-H adsorption/desorption and Pt-O formation/reduction. Unfortunately, these characteristic peaks of Pt cannot be detected on single-atom Pt (Supplementary Fig. 8), owing to the discrete distribution or ultralow loading of Pt atoms [*ACS Catal.* 2017, 7, 3121; *Nat. Commun.* 2019, 10, 1743; *Nat. Catal.* 2018, 1, 985; *Nat. Commun.* 2020, 11, 4558]. This impedes the experimental exploration and measurement of the affinity of single-atom Pt towards H/OH species. Exploiting new experimental tools and techniques to quantify the interaction between single-atom Pt and OH_{ad} is very essential but beyond the scope of the current work.

9. To compare the HER performance of Pt-SAs/TMDs in this work, the authors should cite several relative references based on Pt catalyst in Table 4 and 5.

Response: We thank the reviewer for this valuable advice. We have listed several relative references based on state-of-the-art Pt-based catalysts for alkaline and acidic HER in Supplementary Tabs. 4 and 7.

Supplementary Table 4. Comparison of HER activity for Pt-SAs/MoSe₂ in alkaline solution with the state-of-the-art Pt-based catalysts reported previously.

Catalyst	Loading ($\mu\text{g}_{\text{Pt}} \text{ cm}_{\text{disk}}^{-2}$)	Electrolyte	Current density (mA cm^{-2})	η_{10} (mV)	Tafel slope (mV dec^{-1})	Reference
Pt-SAs/MoSe ₂	2.89	1 M KOH	99.47@100 mV	29	34	This work
BPed-Pt/GR	14.28	1 M KOH	154.7@100 mV	21	46.9	Angew. Chem. Int. Ed., 2019, 58, 19060
PtRu NCs/BP	14.8	1 M KOH	88.5@70 mV	22	19	ACS Catal. 2019, 9, 10870
Pt _{6.2} Ni-S NWs	15.31	1 M KOH	75.3@70 mV	24	NA	Angew. Chem. Int. Ed. 2018, 57, 11678
Pt ₃ Ni ₃ -NWs	15	1 M KOH	39.7@70 mV	40	NA	Angew. Chem. 2016, 128, 13051
SA In-Pt NWs/C	5.1	1 M KOH	~30@90 mV	46	32.4	Adv. Funct. Mater. 2020, 30, 2004310
Ni(OH) ₂ modified Pt	NA	0.1 M KOH	~9.5	~75	75±5	Angew. Chem., Int. Ed., 2012, 51, 12495
PtNi-O/C	5.1	1 M KOH	36.9@70 mV	40	78.8	J. Am. Chem. Soc. 2018, 140, 9046
Pt-Ni ASs	17	1 M KOH	~42@70 mV	27.7	27	Adv. Mater. 2018, 30, 1801741
Pt ₃ Ni ₂ /NiS _x NWs	15.31	1 M KOH	37.2@70 mV	42	NA	Nat. Commun. 2017, 8, 14580
Pt-Co(OH) ₂ /CC	390	1 M KOH	~70@100 mV	32	70	ACS Catal. 2017, 7, 7131
PtCo-Co/TiM	43	1 M KOH	46.5@70 mV	28	35	Nanoscale, 2018, 10, 12302

Ni ₃ N/Pt	~300	1 M KOH	~35@70 mV	50	36.5	Adv. Energy Mater. 2016, 1601390
PtNi–Ni NA/CC	69.3	0.1 M KOH	~20@60 mV	38	42	Inorg. Chem. Front., 2018, 5, 1365
hcp -Pt-Ni	8	0.1 M KOH	~24.2@70 mV	~67	74	Nat. Commun., 2017, 8, 15131
Ni(OH) ₂ /Pt(111) surface	NA	0.1 M KOH	~2.2	~138	~100-130	Science, 2011, 334, 1256
Pt NWs/SL-Ni(OH) ₂	16	1 M KOH	10.9	~70	NA	Nat. Commun. 2015, 6, 6430
		0.1 M KOH	26.6	~48	NA	

Supplementary Table 7. Comparison of HER activity for Pt-SAs/WS₂ in acidic solution with the state-of-the-art Pt-based catalysts reported previously.

Catalyst	Loading ($\mu\text{g}_{\text{Pt}} \text{ cm}_{\text{disk}}^{-2}$)	Electrolyte	Current density (mA cm ⁻²)	η_{10} (mV)	Tafel slope (mV dec ⁻¹)	Reference
Pt-SAs/WS ₂	0.415	0.5 M H ₂ SO ₄	54@100 mV	32	28	This work
PtW NPs/C	20.38	0.5 M H ₂ SO ₄	NA	19.4	27.8	J. Am. Chem. Soc. 2020, 142, 17250
PtFeCo	51	0.5 M H ₂ SO ₄	1325@400 mV	NA	21	Adv. Mater. 2016, 28, 2077
Pt/TiO ₂	0.21 (μg_{Pt})	0.5 M H ₂ SO ₄	7@100 mV	121	40	Energ. Environ. Sci. 2017, 10, 2450
CuPdPt/C	0.27	0.5 M H ₂ SO ₄	~120@100 mV	55	25	J. Mater. Chem. A, 2016, 4, 15309
β -Ni ₂ P ₂ O ₇ /Pt	1	0.5 M H ₂ SO ₄	30@50 mV	28	32	ACS Appl. Mater. Interfaces 2019, 11, 4969
AuPt NDs	47.5	0.5 M H ₂ SO ₄	~34@70 mV	50	34	Int. J. Hydrog. Energy 2016, 42, 18193.

Pt/def- WO ₃ @CFC	15.9	0.5 M H ₂ SO ₄	~28@100 mV	42	61	J. Mater. Chem. A, 2019, 7, 6285
Pt ₅₃ Ru ₃₉ Ni ₈	NA	0.5 M H ₂ SO ₄	NA	37	34	J. Colloid Interface Sci. 2017, 505, 14
PtAg NCs	59.8	0.5 M H ₂ SO ₄	100@150 mV	~60	40	J. Colloid Interface Sci. 2017, 494, 15
Pt NP/N- graphene	5.65	0.5 M H ₂ SO ₄	24@50 mV	30	28	Nanoscale, 2017, 9, 10138
Au _{38.4} @Au _{9.3} Pt _{52.3} -NP/C	17.16	0.5 M H ₂ SO ₄	49.1@40 mV	16	14	Nanoscale, 2020, 12, 20456
10Pt@HN-BC	12	0.5 M H ₂ SO ₄	~30@60 mV	47	35	Int. J. Hydrogen Energy 2018, 43, 6167
H-PtNiCu- AAT	26.5	0.1 M HClO ₄	~70@100 mV	32	33	ACS Appl. Mater. Interfaces 2020, 12, 9600
PtCoFe@CN	13	0.5 M H ₂ SO ₄	~16@50 mV	45	32	ACS. Appl. Mater. Interface 2017, 9, 3596
Pt ₇₅ Mo ₂₅ /rGO	76	0.5 M H ₂ SO ₄	~65@100 mV	32	32	Carbon 2019, 146, 116
PtPd@NSL	19.7	0.5 M H ₂ SO ₄	~55@50 mV	29	23	J. Electroanal. Chem. 2018, 822, 10
80Pt/C-MOF	24.4	0.5 M H ₂ SO ₄	50@50 mV	42.1	24.45	J. Mater. Chem. A 2019, 7, 20239

Reviewer #2:

In this work, the author reported that the single-atom Pt catalysts are located on TMDs supports (MoS_2 , WS_2 , MoSe_2 , and WSe_2). The oxidation state of monoatomic Pt controls the catalytic activity of HER by adjusting the Pt–H/Pt–OH interaction. There is a very large amount of experimental data. Thus, I think this manuscript meet the high criteria of Nature Communications and recommend its publication after mini revision. The following comments should be taken for more consideration.

Response: We thank this reviewer for his/her positive comments. The suggestions are valuable and constructive to helping us to enhance the quality of our manuscript. We have very carefully revised the manuscript and replies to the comments point-by-point.

1. It would be better if the TEM of the catalysts were provided after 1000 cycles stability tests.

Response: We thank the reviewer for this constructive advice. TEM characterization of Pt-SAs/TMDs after HER measurements suggested the unchanged morphology of single-atom Pt during HER, further confirming the stability of Pt-SAs/TMDs catalysts. Relevant description and figure have been added in the revised manuscript (changes are highlighted) and supplementary materials (Supplementary Fig. 23).

Supplementary Figure 23. Atomic-resolution HAADF-STEM images for the Pt-SAs/TMDs samples after HER measurements (scale bars: 2 nm).

2. The format for the references is not uniform, please check.

Response: We thank the reviewer for this correction. We have carefully double-checked the format of all the references and made it uniform throughout the paper.

3. In this work, the authors tune the electronic metal–support interaction by change the support, for example, MoS₂, WS₂, MoSe₂, and WSe₂. If the authors try to tune the electronic metal–support interaction (such as the local coordination environment of single-atom) on the same support, some related works about this issue can be refereed, for example, Modulating the local coordination environment of single-atom catalysts for enhanced catalytic performance, *Nano Res.*, 2020, 13, 1842–1855; Atomic-Level Modulation of Electronic Density at Cobalt Single-Atom Sites Derived from Metal–Organic Frameworks: Enhanced Oxygen Reduction Performance, *Angew. Chem. Int. Ed.*, 2020, DOI: 10.1002/anie.202012798; etc.

Response: We thank this reviewer for the valuable suggestion. We have added the related works in Refs. 38–41. These works offer us deeper atomic-level insights into the relationship between electronic density/coordination environment of active site and catalytic performances, promoting rational design of highly efficient catalysts. [Modulating the local coordination environment of single-atom catalysts for enhanced catalytic performance, *Nano Res.*, 2020, 13, 1842; Atomic-Level Modulation of Electronic Density at Cobalt Single-Atom Sites Derived from Metal–Organic Frameworks: Enhanced Oxygen Reduction Performance, *Angew. Chem. Int. Ed.*, 2021, 60, 3212; In Situ Phosphatizing of Triphenylphosphine Encapsulated within Metal–Organic Frameworks to Design Atomic Co₁–P₁N₃ Interfacial Structure for Promoting Catalytic Performance, *J. Am. Chem. Soc.* 2020, 142, 8431; Tuning the Coordination Environment in Single-Atom Catalysts to Achieve Highly Efficient Oxygen Reduction Reactions, *J. Am. Chem. Soc.* 2019, 141, 20118].

4. More discussion is needed to emphasize the advantages of the EMSI effect in the structure and activity relationship section.

Response: We thank the reviewer for this constructive and valuable advice. We have added more discussion to emphasize the advantages of the EMSI effect in the structure and activity relationship section (Discussion section in the manuscript):

“In this work, we demonstrate that EMSI modulation of single-atom Pt significantly regulates the HER activity over a wide pH range, and systematically unravel the relationship between oxidation state and HER activity of single-atom Pt. The EMSI—acting as a bridge between electronic study and catalyst design—provides a detailed explanation of the enhanced properties of supported catalysts at the electronic scale. With the length scale of catalysts shrinking to atomic level, the EMSI effect becomes stronger and can predominate the reaction rate [*Adv. Mater.* 2020, 32, 2003300; *Nano Res.*, 2020, 13, 1842]. The strong EMSI between the single-atom Pt and TMDs support redistributes the electron density around the metal center with the direct formation of metal–support bonds, facilitating the electron transfer from the active metal center to the reactant. The changes in the oxidation state of single-atom Pt can be the direct effect from EMSI, which acts as a useful approach to quantitatively determine the strength of EMSI. The technical characterizations (*e.g.* XPS, XAS; Fig. 2) of the oxidation state pave the way for revealing the underlying mechanism of the target reaction, which in turn enhances the comprehensive understanding of EMSI and electronic structures across the length scales. Apart from changes in the electronic structure of active sites, the stabilization effect is also a basic influence of EMSI, which suppresses the migration of single-atom metals even under operating conditions (Supplementary Figs. 23 and 24).”

Reviewer #3:

Although the studied systems and results are very interesting and the methodology combined approaches able to illuminate a microscopic picture of investigated catalysts, unfortunately, I can not recommend the publication of the paper due to significant similarity to the Ref. 31 - "Site-specific electrodeposition enables self-terminating growth of atomically dispersed metal catalysts" - recently published work of several co-authors of the present manuscript (Nature Communication 11, 4558 (2020)).

In Ref. 31, the authors mainly studied Pt atoms at MoS₂ but also included results for Pt at WS₂ and MoSe₂, as well as a few other metals at MoS₂. In this paper, they focused on Pt-based catalysts. In addition to three substrates considered in Ref. 31, WSe₂ was included as the 4th support.

In the present paper the main results, highlighted in the abstract, are increased activity of Pt₁/WS₂ and Pt₁/MoSe₂, compared to Pt/C and the correlation of the activity and the Pt interaction with the support. However, the activity of several of the inspected catalysts was already compared in Fig. 24, Supported Information (SI) of Ref. 31. In Table 4, SI of Ref. 31 were also presented DFT results for the binding energies of Pt adatom at MoS₂, WS₂ and MoSe₂ (these results are partially presented again in Table 3, the SI of the present manuscript).

Due to apparent similarity in studied systems, methodology, and results, the present manuscript could be considered as a follow up to Ref. 31, and thus lacks sufficient novelty compulsory for the publication in Nature Communications.

Response: We thank the reviewer for the comments. We found that we didn't clearly convey the information that precisely distinguish our current work with previous work described in ref. 31. We provided a detailed explanation here and revised our manuscript accordingly.

The previous work (Ref. 31) developed a site-specific electrodeposition method for synthesis of single-atom materials and focused on **the growth mechanism** of single-atom metals. Different from the main idea of **previous work (single-atom growth)**, **the aim of the current work is to unveil the mechanism of wide-pH-range HER and mechanistic structure–activity relationship for catalysts design**. We used many different advanced spectroscopies (UHV-XPS, UPS, XAS and NAP-XPS) and electrochemical techniques to characterize the EMSI-induced changes in the electronic structure of single-atom and its corresponding alkaline and acidic HER activities.

Our current work used experimental techniques to explore and quantify the EMSI effects on the wide-pH-range HER at the atomic level, which was realized by correlating the acidic/alkaline HER activities with the average oxidation state of single-atom Pt and the Pt–H/Pt–OH interaction. These results help to understand the HER mechanism and satisfies the high demand for deep understanding of structure–activity relationship. **We show that one descriptor (H^*) governs the acidic HER while two descriptors (OH^* and H^*) co-determined the rate of alkaline HER, and the structure–activity relationship can be used as a universal guideline for the design of single-atom metal catalysts.**

In view of the reviewer’s concern, to better highlight the concept of the current work, we revised the paper from the following aspects:

(1) We have revised the abstract and emphasized the EMSI modulation to avoid the confusion of the main results: “Here, we reveal that the fine control over the oxidation states of single-atom Pt catalysts through electronic metal–support interaction significantly modulates the catalytic activities in either acidic or alkaline HER”.

(2) We have also cited the ref. 31 to the investigation of acidic HER, which makes our description more precise. “Similarly, we also showed the various acidic HER activities

on different supported single-atom Pt samples, which followed the order distinctive from that in alkaline HER: Pt-SAs/WS₂ > Pt-SAs/MoS₂ > Pt-SAs/MoSe₂ > Pt-SAs/WSe₂ (Fig. 3e and 3f), in consistence with the previously reported results [*Nat. Commun.* 2020, 11, 4558].”

(3) Following the reviewer’s valuable suggestion, we replaced Supplementary Tab. 3 with the energy barriers for Pt diffusion between the nearest adsorption sites to clarify the structural stability of single-atom Pt. The updated and optimized DFT calculation has been added in Supplementary Material (Supplementary Fig. 16).

Supplementary Figure 16. Energy as a function of reaction coordinate for the diffusion of single-atom Pt between the nearest adsorption sites on MoS₂. Model I represents the most stable structure for Pt-SAs/MoS₂, while model II~V represent the intermediate metastable states for Pt-SAs/MoS₂.

(4) Emphasizing the importance of the structure–activity relationship (reported in this work) as a universal guideline for the design of single-atom metal catalysts (Discussion section in Manuscript):

“From the structure–activity relationship, the fine control over the oxidation state of single-atom Pt catalysts enables to achieve the optimal catalytic activity in either acidic or alkaline HER. In acidic environment, the HER performance could be well optimized through properly decreasing the oxidation state of single-atom Pt (Fig. 5a), accelerating the hydrogen desorption process. Similar to our finding, Liang’s and Yao’s groups have recently reported that the electron-enriched or near-zero-valence atomically dispersed Pt species are more active than the high-valence single-atom Pt for catalyzing acidic HER [*Nat. Commun.* 2020, 11, 1029; *Nat. Commun.* 2019, 10, 4977]. In alkaline environment, as the oxidation state of single-atom Pt increases, the HER activity also increases initially (left side of the volcano plot, Fig. 4e). Under the electrochemical condition, the charge of the metal plays a decisive role with regard to water dissociation [*Nat. Energ.* 2017, 2, 17031; *J. Electroanal. Chem.* 1996, 411, 95]. Choi’s group reported the counterintuitive promoting effect of CO molecule on alkaline HER of single-atom Pt [*J. Am. Chem. Soc.* 2018, 140, 16198], which could be also explained by our proposed model (Fig. 5b). After the coordination of CO (strong electron acceptor), metal-to-ligand charge transfer from Pt was increased, endowing the single-atom Pt site with higher oxidation state (*ca.* +2–2.3). Such single-atom Pt site was more electrophilic and favorable for the water dissociation, thus accelerating the alkaline HER. The opposite trends in acidic and alkaline HER indicate the different mechanistic pathways of HER depending on the pH conditions, consistent with the previously reported works [*J. Am. Chem. Soc.* 2018, 140, 16198; *Nat. Mater.* 2016, 15, 197].

We further suggest that single-atom Pt with optimal valence state is simultaneously favourable for water dissociation, adsorption/desorption of OH* and H* (Fig. 5b, and

Supplementary Fig. 34). It should be noted that with the increased oxidation state of single-atom Pt, the adsorption of H^* and OH^* on the catalyst were both strengthened. Although single-atom Pt with high oxidation state energetically favors the adsorption of electron-rich H_2O and OH^* , too strong hydrogen adsorption will also lead to the slow release of active sites and hence the sluggish HER kinetics. From a fundamental point of view, the present work shows the atomic-level enhancement of HER thermodynamically and kinetically by optimizing the catalyst–H interaction and accelerating the water dissociation. The single-atom Pt catalyst with optimal oxidation state ($\sim+2$), showed neither too strong Pt–H interaction to release hydrogen, nor too weak Pt–OH interaction to dissociate water, which dramatically contributes to the overall alkaline HER.”

(5) More description and discussion about the debate on HER mechanism have been added (Discussion section in Manuscript):

“To date, three widely adopted theories have emerged to explain the alkaline HER mechanism: water dissociation theory (hydroxyl binding energy, OHBE), hydrogen binding energy (HBE) theory, or interface water and/or anion transfer theory [*Mater. Today* 2020, 36, 125]. It still remains unclear which descriptor governs the alkaline HER. At the atomic level, our results show that the two descriptors (OH^* and H^*) co-determined the rate of alkaline HER (Fig. 5b), which sheds light on the long-standing puzzle about HER mechanism.”

The present manuscript, which I consider as a nice work, could be improved by including a better description of the Pt interaction with TMDs. The effect of the support should be presented in more detail. These details can be easily obtained from DFT calculations, already performed by the authors:

1. In Fig. 4a are shown positions of *d*-bands from UPS valence-band spectra. The authors should include DOS projected on Pt-*d* orbitals and the *d*-states occupation numbers, calculated using DFT.

Response: We thank the reviewer for this valuable advice. “Due to the EMSI modulation, various density of states patterns were obtained from the DFT calculation for the Pt-*d* orbitals of the four Pt-SAs/TMDs (Supplementary Fig. 28)”. Relevant description and figures have been added in the manuscript and supplementary material (Supplementary Fig. 28).

The DFT-calculated positions of *d*-band center follows the order: Pt-SAs/MoS₂ (-8.10) < Pt-SAs/WS₂ (-7.96) < Pt-SAs/MoSe₂ (-7.44) < Pt-SAs/WSe₂ (-7.39). Note that the experimental results show the positions of *d*-band center with the order of Pt-SAs/WS₂ < Pt-SAs/MoS₂ < Pt-SAs/MoSe₂ < Pt-SAs/WSe₂, whose overall trend is consistent with the theoretical results. The only discrepancy lies in the higher position of Pt-SAs/MoS₂. We considered that the discrepancy is mainly caused by the approximations adopted in our calculations for achieving the compromise between accuracy and affordable computational cost. First, the TMD systems used in calculation are simplified to the perfect surface and different from the real situation in the experimental study. Second, the dipole of the whole system is not considered owing to that only one Pt atom was involved in the calculation. Nevertheless, the DFT-calculated results still rationalize the experimental trends (Pt-SAs/WS₂ or Pt-SAs/MoS₂ < Pt-SAs/MoSe₂ < Pt-SAs/WSe₂), which is of great reference value to our work.

Supplementary Figure 28. Density of states projected on Pt-*d* orbitals of Pt-SAs/TMDs. The white dashed lines represent the *d*-band center calculated by DFT.

2. In Table 3 of the SI are listed Pt binding energies. Even more useful information would be the energy barriers for Pt diffusion between the nearest adsorption sites. The calculated activation energies will demonstrate the structural stability of the studied single-atom catalysts. This is an important issue, studied by Li et al. (ACS Nano 11, 3392 (2017)), where they claimed that Pt atoms at the ideal 2H-MoS₂ were very mobile in the room temperature and only those trapped at the S-vacancies were structurally stable. In the present study, one of the supports was 1T phase of MoS₂, so the picture might be different from 2H-MoS₂, but the authors should clarify this by carrying out the corresponding DFT calculations.

Response: We thank the reviewer for this question and valuable suggestions. We further clarified the structural stability of the single-atom catalysts with the information about energy barriers for Pt diffusion between the nearest adsorption sites from DFT calculation. Relevant description and figures have been added in the manuscript and Supplementary Material: “The high energy barrier for Pt diffusion between the nearest adsorption sites further demonstrated the structural stability of the single-atom catalysts (Supplementary Fig. 16)”.

We have calculated the activation energy of the Pt atom diffusion on the 1T-MoS₂ along two directions. The energy barrier of the Pt atom diffusion between the nearest adsorption sites is 1.0-1.1 eV (Supplementary Fig. 16), which is higher than the value (0.6-0.8 eV) on 2H-MoS₂ reported by Li et al. [ACS Nano 11, 3392 (2017)]. Following the Arrhenius relation, the high barrier by 0.4 eV suggests significantly enhanced stability at room temperature.

Supplementary Figure 16. Energy as a function of reaction coordinate for the diffusion of single-atom Pt between the nearest adsorption sites on MoS₂. Model I represents the most stable structure for Pt-SAs/MoS₂, while model II~V represent the intermediate metastable states for Pt-SAs/MoS₂.

REVIEWERS' COMMENTS

Reviewer #1 (Remarks to the Author):

I am satisfied with the revision, and it can be accepted at the present state.

Reviewer #2 (Remarks to the Author):

The revised version can be accepted.

Reviewer #3 (Remarks to the Author):

I am very pleased with the authors' answer to my comments and the corresponding changes brought to the manuscript.

In the revised paper and the response to my comments, they emphasized differences between this work and former Ref. 31 (now Ref. 43), particularly detailed description of the mechanism behind favorable catalytic properties of the single-atom metal catalysts over a wide-pH-range. Since they applied the same methods in two papers, it is difficult to avoid the similarity in the Figures with results, and it is me who did not see this very clear when reading the original version of the paper.

Therefore, I recommend the publication of the paper in the present form.

A minor comment to Figure 4c and 4e:

The y-axis in the graph in Figure 4c,d labeled as "H/OH adsorption ability" should include a quantity that can be measured or calculated - for example, d-band center. The "adsorption ability" is a loose term.

REVIEWERS' COMMENTS:

Reviewer #1 (Remarks to the Author):

I am satisfied with the revision, and it can be accepted at the present state.

Response: We thank the reviewer for his/her positive comments.

Reviewer #2 (Remarks to the Author):

The revised version can be accepted.

Response: We thank the reviewer for his/her positive comments.

Reviewer #3 (Remarks to the Author):

I am very pleased with the authors' answer to my comments and the corresponding changes brought to the manuscript. In the revised paper and the response to my comments, they emphasized differences between this work and former Ref. 31 (now Ref. 43), particularly detailed description of the mechanism behind favorable catalytic properties of the single-atom metal catalysts over a wide-pH-range. Since they applied the same methods in two papers, it is difficult to avoid the similarity in the Figures with results, and it is me who did not see this very clear when reading the original version of the paper. Therefore, I recommend the publication of the paper in the present form.

A minor comment to Figure 4c and 4e: The y-axis in the graph in Figure 4c,d labeled as "H/OH adsorption ability" should include a quantity that can be measured or calculated - for example, d-band center. The "adsorption ability" is a loose term.

Response: We thank the reviewer for this minor comment. The H adsorption ability is quantified by the position of the *d*-band center (Fig. 4a), while the Pt–OH interaction is quantified by the CO oxidation potentials obtained from CO stripping voltammetry (Fig. 4d). We have highlighted the relevant description in the Figure legend.

Fig. 4 | Mechanistic investigations. (a) UPS valence-band spectra (VBS) of single-atom Pt relative to the Fermi level. The black dashed lines represent the position of the d -band centers. The grey dashed lines represent the variation in the VBS of ce-TMDs induced by single-atom Pt attachment. (b) Schematic DOS diagrams illustrating the EMSI effect on the d -band position of single-atom Pt, and the interaction between Pt and chemisorbed atomic hydrogen. When H is adsorbed on single-atom Pt, the interaction of the adsorbed H (H^*) s -orbital with the Pt d -orbital will result in fully filled low-energy bonding states and partially filled high-energy antibonding states. (c) Relationship of average oxidation state, H adsorption ability and acidic HER activity of Pt-SAs/TMDs. The circle and triangle represent the average oxidation state of single-atom Pt obtained from XPS and XANES, respectively. **The H adsorption ability is quantified by the position of the d -band center.** (d) CO stripping voltammetry of various single-atom Pt catalysts. The dashed lines represent the CO oxidation potentials obtained for catalysts. The scanning potential value for Pt-SAs/TMDs samples could only reach as high as 0.7 V, since higher potential resulted in the oxidation of ce-TMDs (Supplementary Fig. 36). (e) Relationship of average oxidation state, Pt–OH interaction and alkaline HER activity of Pt-SAs/TMDs. The circle and triangle are the average oxidation states of single-atom Pt obtained from XPS and XANES, respectively. **The Pt–OH interaction is quantified by the CO oxidation potentials obtained from CO stripping voltammetry.**